# Anti-diabetic drug binding site in a mammalian $K_{ATP}$ channel revealed by Cryo-EM

Gregory M Martin[1], Balamurugan Kandasamy[1], Frank DiMaio[2], Craig Yoshioka[3]*, Show-Ling Shyng[1]*

[1]Department of Biochemistry and Molecular Biology, Oregon Health and Science University, Portland, United States; [2]Department of Biochemistry, University of Washington, Seattle, United States; [3]Department of Biomedical Engineering, Oregon Health and Science University, Portland, United States

**Abstract** Sulfonylureas are anti-diabetic medications that act by inhibiting pancreatic $K_{ATP}$ channels composed of SUR1 and Kir6.2. The mechanism by which these drugs interact with and inhibit the channel has been extensively investigated, yet it remains unclear where the drug binding pocket resides. Here, we present a cryo-EM structure of a hamster SUR1/rat Kir6.2 channel bound to a high-affinity sulfonylurea drug glibenclamide and ATP at 3.63 Å resolution, which reveals unprecedented details of the ATP and glibenclamide binding sites. Importantly, the structure shows for the first time that glibenclamide is lodged in the transmembrane bundle of the SUR1-ABC core connected to the first nucleotide binding domain near the inner leaflet of the lipid bilayer. Mutation of residues predicted to interact with glibenclamide in our model led to reduced sensitivity to glibenclamide. Our structure provides novel mechanistic insights of how sulfonylureas and ATP interact with the $K_{ATP}$ channel complex to inhibit channel activity.

DOI: https://doi.org/10.7554/eLife.31054.001

*For correspondence:
yoshiokc@ohsu.edu (CY);
shyngs@ohsu.edu (S-LS)

Competing interests: The authors declare that no competing interests exist.

## Introduction

ATP-sensitive potassium ($K_{ATP}$) channels are unique hetero-octameric complexes each composed of four inwardly rectifying Kir6 channel subunits and four sulfonylurea receptor (SUR) subunits belonging to the ATP binding cassette (ABC) transporter protein family (*Aguilar-Bryan and Bryan, 1999*; *Nichols, 2006*). In pancreatic β-cells, $K_{ATP}$ channels formed by Kir6.2 and SUR1 are gated by intracellular ATP and ADP, with ATP inhibiting channel activity while $Mg^{2+}$-complexed ATP and ADP stimulating channel activity (*Aguilar-Bryan and Bryan, 1999*; *Ashcroft, 2007*). During glucose stimulation, the intracellular ATP to ADP ratio increases following glucose metabolism, which favors channel closure by ATP, resulting in membrane depolarization, $Ca^{2+}$ influx, and exocytosis of insulin granules. In this way, $K_{ATP}$ channels are able to control insulin secretion according to blood glucose levels. Mutations that disrupt channel function are known to cause a spectrum of insulin secretion disorders (*Ashcroft, 2005*; *Koster et al., 2005a*). Specifically, loss-of-function mutations result in congenital hyperinsulinism, whereas gain-of-function mutations lead to transient or permanent neonatal diabetes (*Ashcroft, 2005*). The pivotal role of $K_{ATP}$ channels in insulin secretion regulation makes them an important drug target.

Discovered in the 1940s, sulfonylureas have been a mainstay of type 2 diabetes therapy for more than half a century (*Sola et al., 2015*). The medical importance of this class of drugs has led to its evolution into several generations of agents, including first-generation sulfonylureas such as tolbutamide and second-generation agents such as the high-affinity sulfonylurea glibenclamide (GBC) (*Gribble and Reimann, 2003*; *Sola et al., 2015*). All sulfonylureas stimulate insulin secretion to

reduce plasma glucose levels by inhibiting the activity of β-cell $K_{ATP}$ channels (*Gribble and Reimann, 2003*). More recently, they have also become the primary pharmacotherapy for neonatal diabetes patients carrying gain-of-function $K_{ATP}$ channel mutations (*Aguilar-Bryan and Bryan, 2008*; *Ashcroft, 2007*; *Sagen et al., 2004*). Despite their clinical importance and decades of research, how sulfonylureas interact with and inhibit $K_{ATP}$ channel activity remains poorly understood.

To begin to address the structural mechanisms by which ATP and sulfonylureas such as GBC inhibit $K_{ATP}$ channels to stimulate insulin secretion, we recently carried out single particle cryo-EM and determined the structure of the β-cell $K_{ATP}$ channel complex in the presence of ATP and GBC (*Martin et al., 2017*). While our initial structure at a resolution of 5.7 Å revealed the overall architecture of the channel and location of the ATP molecule, it was unable to clearly define the GBC binding site and the atomic details associated with ATP binding. A concurrent study by Li *et al.* (*Li et al., 2017*) reported another cryo-EM $K_{ATP}$ channel structure at 5.6 Å resolution, also in the presence of GBC but without ATP, in which the GBC binding site was proposed to lie near the cytoplasmic linker between the first and second transmembrane domains of SUR1; however, the assignment of the GBC density was tentative. To resolve the binding sites for ATP and GBC, we performed additional studies and improved the resolution of the $K_{ATP}$ channel structure bound to GBC and ATP to ~3.6 Å. The higher resolution structure not only clearly defines the GBC and ATP binding pockets but also provides novel insights into the mechanisms of channel inhibition by ATP and GBC.

## Results

### Structure determination

To obtain a structure of $K_{ATP}$ channels bound to GBC and ATP, channels comprising a rat Kir6.2 and FLAG-tagged hamster SUR1 (96% and 95% sequence identity to human sequences, respectively) were expressed in rat insulinoma INS-1 832/13 cells (*Hohmeier et al., 2000*), affinity purified, and imaged in the presence of 1 mM ATP (no $Mg^{2+}$) and 1 μM GBC, as described previously (*Martin et al., 2017*). To improve resolution, we adjusted sample and grid preparation parameters (for details see Materials and methods) to optimize ice thickness and particle orientation distributions, which both increased the overall quality and quantity of single particles.

3D classification in Relion yielded one four-fold symmetric class, which reached an overall resolution of 4.07 Å after refinement (*Figure 1—figure supplement 1*; *Table 1*). Particles from this class were further classified and refined using Frealign (*Grigorieff, 2016*), which yielded a map with improved resolution of 3.63 Å (*Figure 1—figure supplement 1*; *Table 1*). The local resolution, as estimated by Bsoft, varied from 3.2 Å in the Kir6.2 transmembrane domain (TMD) to ~5 Å in the SUR1 nucleotide binding domains (NBDs) (*Figure 1—figure supplement 2*). Overall, the map displays excellent connectivity to allow for model building (*Figure 1*). We have constructed a full atomic model for all of Kir6.2 minus disordered N- and C-termini, and for TMD0 of SUR1, as this part of the map was well resolved, with clear side-chain density for most residues. The ABC core of SUR1 displayed greater variability in resolution: the inner helices (relative to Kir6.2/TMD0) were also very well resolved (between 3.5 and 4 Å resolution) to permit nearly complete atomic model building, while the most exposed helices (TMs 9, 10, 12, and 13) showed signs of flexibility, and were only built as polyalanine chains. This was also the case for the NBDs, for which we only refined our previously-deposited NBD models as rigid bodies (see Materials and methods).

### Structural overview

The $K_{ATP}$ channel is built around a tetrameric Kir6.2 core with each subunit in complex with one SUR1 (*Figure 1B–E*), as observed previously (*Martin et al., 2017*). Each Kir6.2 has the typical Kir channel architecture of an N-terminal cytoplasmic domain, a TMD consisting of two TMs termed M1 and M2 interspersed by a pore loop and selectivity filter, and a 'tether' helix that links the TMD to the larger C-terminal cytoplasmic domain (CTD) (*Figure 2A*, *Figure 2—figure supplement 1*). In our new structure, constrictions in the selectivity filter (T130), bundle crossing (F168), and the G-loop (G295, I296) are clearly seen (*Figure 2B,C*) to indicate a closed pore.

SUR1 is one of only a handful of ABC transporters which possesses an N-terminal transmembrane domain, TMD0, in addition to an ABC core structure comprising two TMDs of 6 helices each and two cytosolic NBDs (*Tusnády et al., 2006*; *Wilkens, 2015*) (*Figure 2D*). In the structure, TMD0 is a

**Table 1.** Statistics of cryo-EM data collection, 3D reconstruction and model building.

| Data collection/processing | |
| --- | --- |
| Microscope | Krios |
| Voltage (kV) | 300 |
| Camera | Gatan K2 Summit |
| Camera mode | Super-resolution |
| Defocus range (µm) | −1.4 ~ −3.0 |
| Movies | 2180 |
| Frames/movie | 60 |
| Exposure time (s) | 15 |
| Dose rate (e⁻/pixel/s) | 8 |
| Magnified pixel size (Å) | 1.72 (Super-resolution pixel size 0.86) |
| Total Dose (e-/Å2) | ~40 |
| Reconstruction | |
| Software | Relion and Frealign |
| Symmetry | C4 |
| Particles refined | 59,417 |
| Resolution (Relion masked) | 4.07 Å |
| Resolution (Frealign masked) | 3.63 Å |
| Model Statistics | |
| MapCC | 0.758 (masked) |
| Clash score | 9.10 |
| Molprobity score | 1.9 |
| Cβ deviations | 0 |
| Ramachandran | |
| Outliers | 0.12% |
| Allowed | 6.31% |
| Favored | 93.57% |
| RMS deviations | |
| Bond length | 0.01 |
| Bond angles | 1.11 |

DOI: https://doi.org/10.7554/eLife.31054.005

well-resolved 5-TM bundle (*Figure 2D*, *Figure 2—figure supplement 2*). A long intracellular loop L0 which tethers TMD0 to the ABC core is found to contain both cytosolic and amphipathic domains (*Figure 2D*, *Figure 2—figure supplement 2*). The C-terminal 2/3 of L0 is homologous to the 'lasso motif' observed in CFTR (*Liu et al., 2017*; *Zhang and Chen, 2016*) and MRP1 (*Johnson and Chen, 2017*), and indeed, the structures are very similar (*Figure 2E*). SUR1 is found in an 'inward-facing' conformation, with NBDs clearly separated and the vestibule formed by TMD1/TMD2 open towards the cytoplasm. As we noted previously, the two TMD-NBDs show a ~15° rotation and ~10 Å horizontal translation relative to each other (*Figure 2F*) (*Martin et al., 2017*). This lack of symmetry is also seen in recently reported CFTR and MRP1 inward-facing structures (*Johnson and Chen, 2017*; *Liu et al., 2017*; *Zhang and Chen, 2016*). The separation between the two NBDs in our structure is similar to that seen in MRP1 bound to its substrate leukotriene C4 (*Johnson and Chen, 2017*)(see *Figure 2F*). Like SUR1 in which only NBD2 is capable of hydrolyzing ATP while NBD1 harbors a degenerate ATPase site, CFTR and MRP1 also have two asymmetric NBDs (*Wilkens, 2015*), suggesting the relative rotation and translation between the two TMD-NBD halves may be a common characteristic of ABC transporters with asymmetric NBDs.

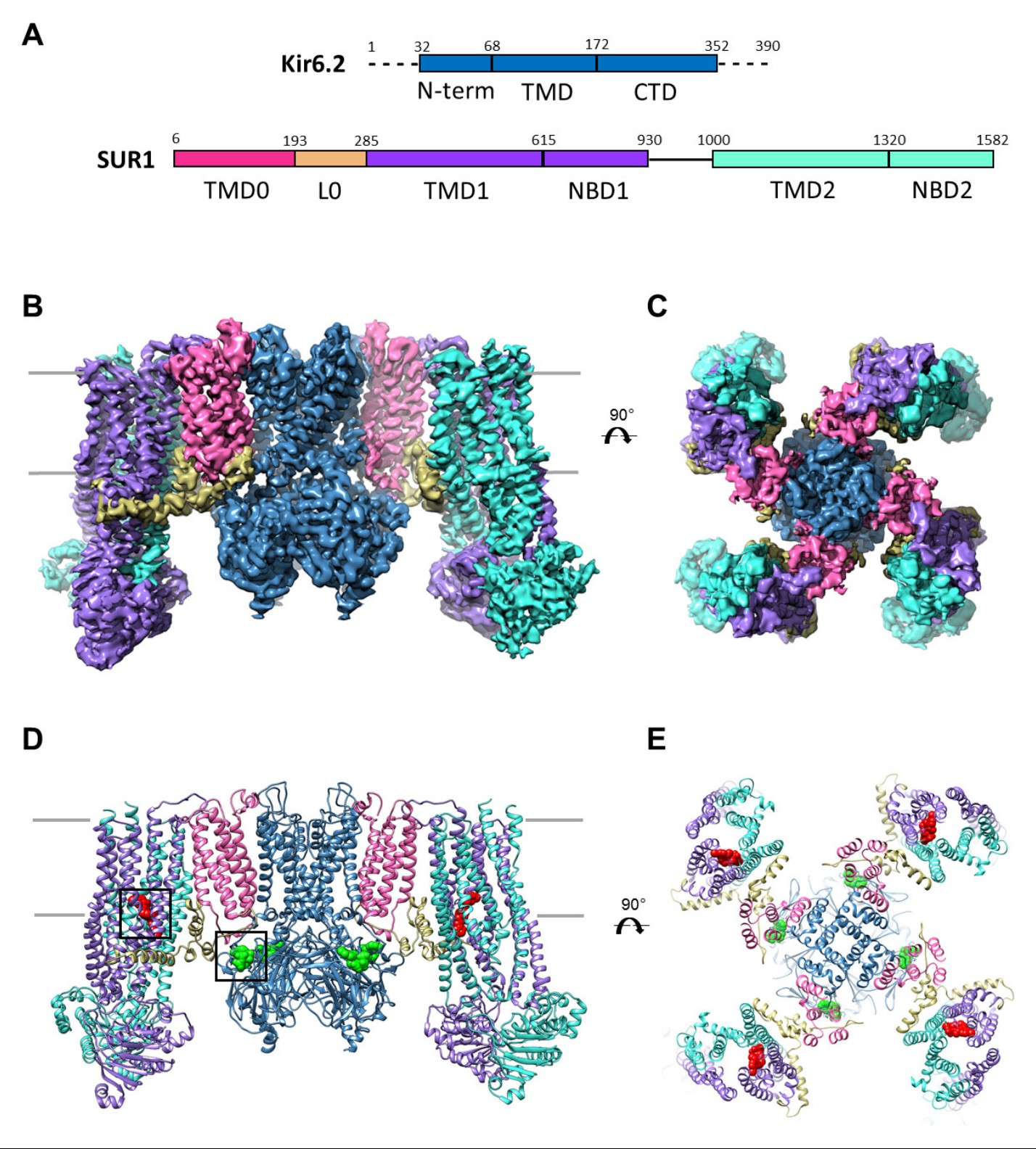

**Figure 1.** Overall structure of the $K_{ATP}$ channel bound to ATP and GBC. (**A**) Linear sequence diagram for the Kir6.2 and SUR1 polypeptides, with primary domains colored to match the panels below. Numbers indicate residue number at the beginning and end of each domain. (**B**) Cryo-EM density map of the $K_{ATP}$ channel complex at 3.63 Å resolution, viewed from the side. Gray bars indicate approximate position of the bilayer. (**C**) View of map from extracellular side. (**D**). Structural model of the complex, with ligands ATP (green) and GBC (red) in boxes. (**E**) View of the model from the extracellular side.

*Figure 1 continued on next page*

*Figure 1 continued*

DOI: https://doi.org/10.7554/eLife.31054.002

The following figure supplements are available for figure 1:

**Figure supplement 1.** Data collection and image processing workflow.

DOI: https://doi.org/10.7554/eLife.31054.003

**Figure supplement 2.** Cryo-EM density map analysis.

DOI: https://doi.org/10.7554/eLife.31054.004

## Unique molecular interactions between SUR1 and Kir6.2

Among all Kir channels, Kir6.1/Kir6.2 are the only members known to couple to an ABC transporter (*Hibino et al., 2010*); and among all ABC transporters, SUR1/SUR2 are the only ones known to couple to an ion channel (*Wilkens, 2015*). These proteins are also unique in that they are co-dependent for both expression and function (*Inagaki et al., 1995*; *Zerangue et al., 1999*). How SUR1 and Kir6.2 achieve this unique regulation has been a long standing question in the field.

In the structure, we find a series of hydrophobic and polar interactions mediated exclusively by TMD0 and L0 of SUR1 with Kir6.2 (*Figure 3*). The extracellular N-terminus of SUR1 closely contacts the turret and pore loop of Kir6.2 (*Figure 3A–D*), while TM1 of TMD0 and the M1 helix of Kir6.2 form a series of hydrophobic interactions running the length of the helices (*Figure 3E*). On the cytoplasmic side, the intracellular loops ICL1, ICL2, and the N-terminal portion of L0 (*Figure 3C and D*), prior to the 'lasso motif,' cluster around the Kir6.2 N-terminus which harbors the slide helix and key ATP-binding residues and also forms part of an intersubunit β-sheet (*Figure 3B and F*).

Of all the Kir channel family members which display a high degree of sequence conservation, only Kir6.1 or Kir6.2 co-assemble with SUR proteins, raising the interesting question of what molecular interactions confer this specificity. While many residue pairs in the interface are conserved in either the Kir or ABC transporter family, a couple residue pairs are unique to both Kir6.2 and SUR1. Among these, H70 within the M1 helix of Kir6.2 forms an edge-to-face π-stacking interaction with W51 of TM1 of TMD0 (*Figure 3E*), and Q57 of the Kir6.2 slide helix contacts F132 of ICL2 (*Figure 3F*). F132 is a well-studied permanent neonatal diabetes mutation which causes very high $P_o$ but also reduces physical interaction between TMD0 and Kir6.2 (*Proks et al., 2007*), supporting its role as a critical part of the interface. To our knowledge, mutational studies of Kir6.2 Q57 and H70, and SUR1 W51 have not been reported; it would be interesting to test the role of these residues in channel assembly and function in the future.

## The ATP binding site

Non-hydrolytic binding of intracellular ATP to the cytoplasmic domains of Kir6.2 induces rapid and reversible closure of the pore (*Nichols, 2006*). We have previously reported the location of the ATP-binding site at the interface of the cytoplasmic N- and C-terminal domains from two adjacent subunits (*Martin et al., 2017*), giving four equivalent sites for the Kir6.2 tetramer. In the current map, there is strong cryo-EM density for the ATP as well as surrounding residues (*Figure 4.*, *Figure 4—figure supplement 1*), allowing for detailed analysis of the mode of ATP binding as well as the possible mechanism of inhibition.

In the structure, the ATP is directly below the inner membrane leaflet and is partially exposed to solvent (*Figure 4*). The bound ATP appears to adopt a conformation similar to that found in other non-canonical, $Mg^{2+}$-independent ATP-binding sites, such as the P2X receptor (*Hattori and Gouaux, 2012*), in which the phosphate groups are folded towards the adenine ring (*Figure 4B,C*). This places the β- and γ-phosphates to interact with basic residues contributed by the N- and C- termini of Kir6.2. The pocket itself is formed by the overlap of three distinct cytoplasmic structures: an N-terminal peptide (binding residues N48 and R50; subunit A) immediately before the Kir channel 'slide helix'; a C-terminal β-sheet (I182 and K185; subunit B) immediately following the TMD-CTD tether helix (see *Figure 2A*, *Figure 2—figure supplement 1*); and a short, solvent-exposed helical segment (Y330, F333, G334; subunit B) (*Figure 4C*). Note an unassigned protruding density close to the ATP density was observed (*Figure 4—figure supplement 1A*). This is reminiscent of a coordinating magnesium ion often observed in other high-resolution structures of proteins bound with ATP or GTP (*Bauer et al., 2000*; *Oliva et al., 2004*). Interestingly, early studies of $K_{ATP}$ channel gating

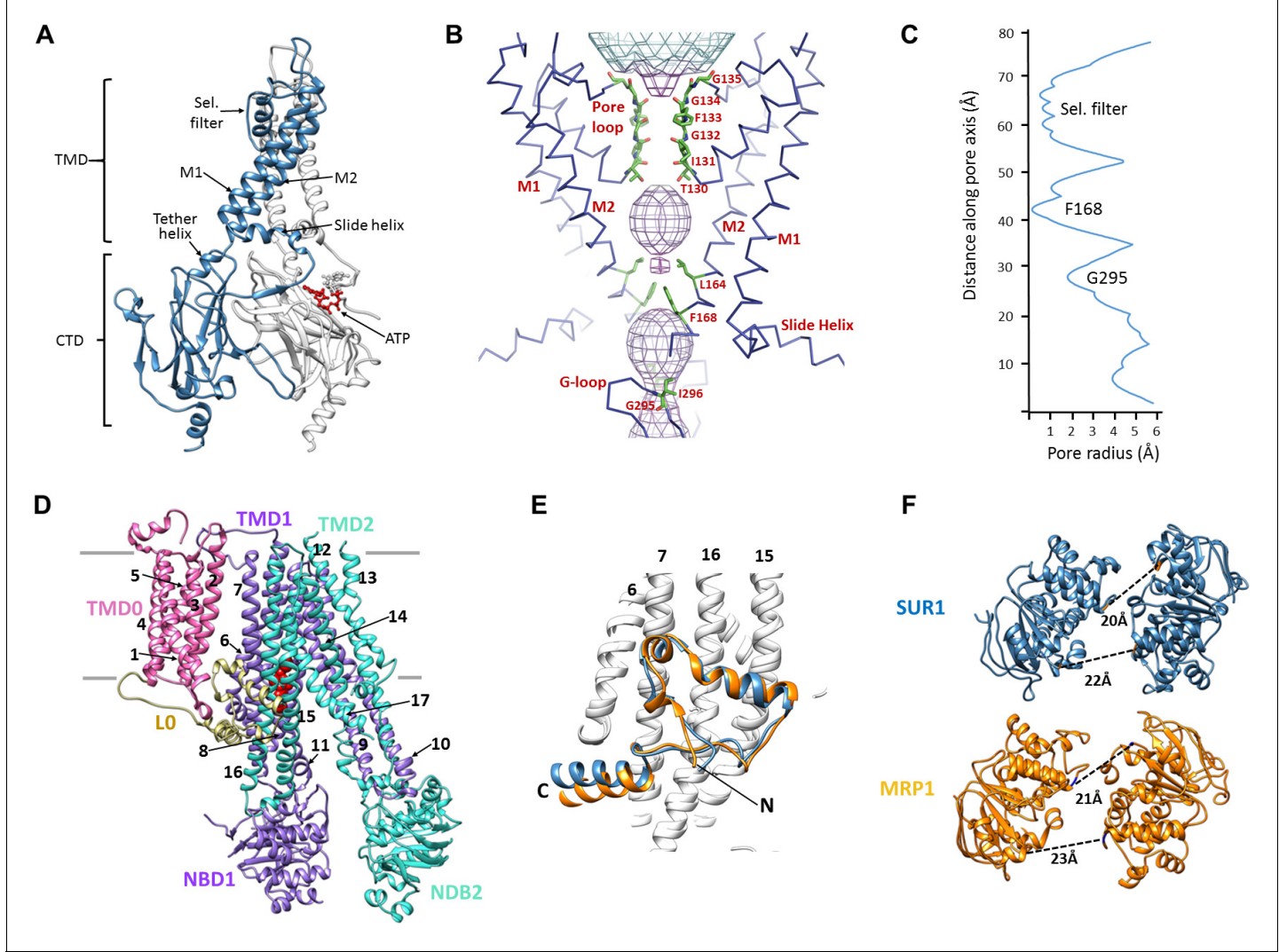

**Figure 2.** Structural highlights of Kir6.2 and SUR1. (**A**) Two subunits of the Kir6.2 tetramer, one colored in blue and one in white, highlighting the conserved Kir channel structural features. Note the ATP-binding site is at the interface of the cytoplasmic N- and C-terminal domains of adjacent subunits. (**B**) Close-up of the Kir6.2 pore, showing solvent-accessible volume as a mesh. The two primary gates are 1) the helix bundle-crossing (HBC), formed by the confluence of the M2 helices at F168; 2) the G-loop, formed at the apex of the CTD by G295 and I296. (**C**) Plot of pore radius as a function of length along pore axis. (**D**) Structure of SUR1 in inward-facing conformation, indicating overall domain organization. Note clear separation of NBDs. Transmembrane helices 1–17 are numbered. (**E**) Structural conservation of L0 with the lasso domain observed in MRP1. Full structures of SUR1 (blue) and leukotriene C4-bound MRP1 (orange) minus TMD0 were used for structural alignment. (**F**) Separation (Cα to Cα, indicated by the dashed line) between Walker A and signature motif in NBD1 (left) and NBD2 (right) (G716::S1483 and S831::G1382 in SUR1, G681::S1430 and S769::G1329 in MRP1).

DOI: https://doi.org/10.7554/eLife.31054.006

The following figure supplements are available for figure 2:

**Figure supplement 1.** Cryo-EM density map of key structural features in Kir6.2.
DOI: https://doi.org/10.7554/eLife.31054.007

**Figure supplement 2.** Cryo-EM density map of transmembrane helices and the lasso (**L0**) motif of SUR1.
DOI: https://doi.org/10.7554/eLife.31054.008

showed that, while $Mg^{2+}$ is not necessary as a cofactor in ATP inhibition, MgATP can inhibit as effectively as free ATP (*Lederer and Nichols, 1989*). Although we did not include $Mg^{2+}$ in our sample, a possibility that low concentrations of $Mg^{2+}$ might be present in the buffer cannot be excluded.

The α-phosphate of ATP is coordinated by the main-chain nitrogen of G334 and K185, while the β- and γ-phosphates are coordinated by side-chain nitrogens of K185 and R50, respectively

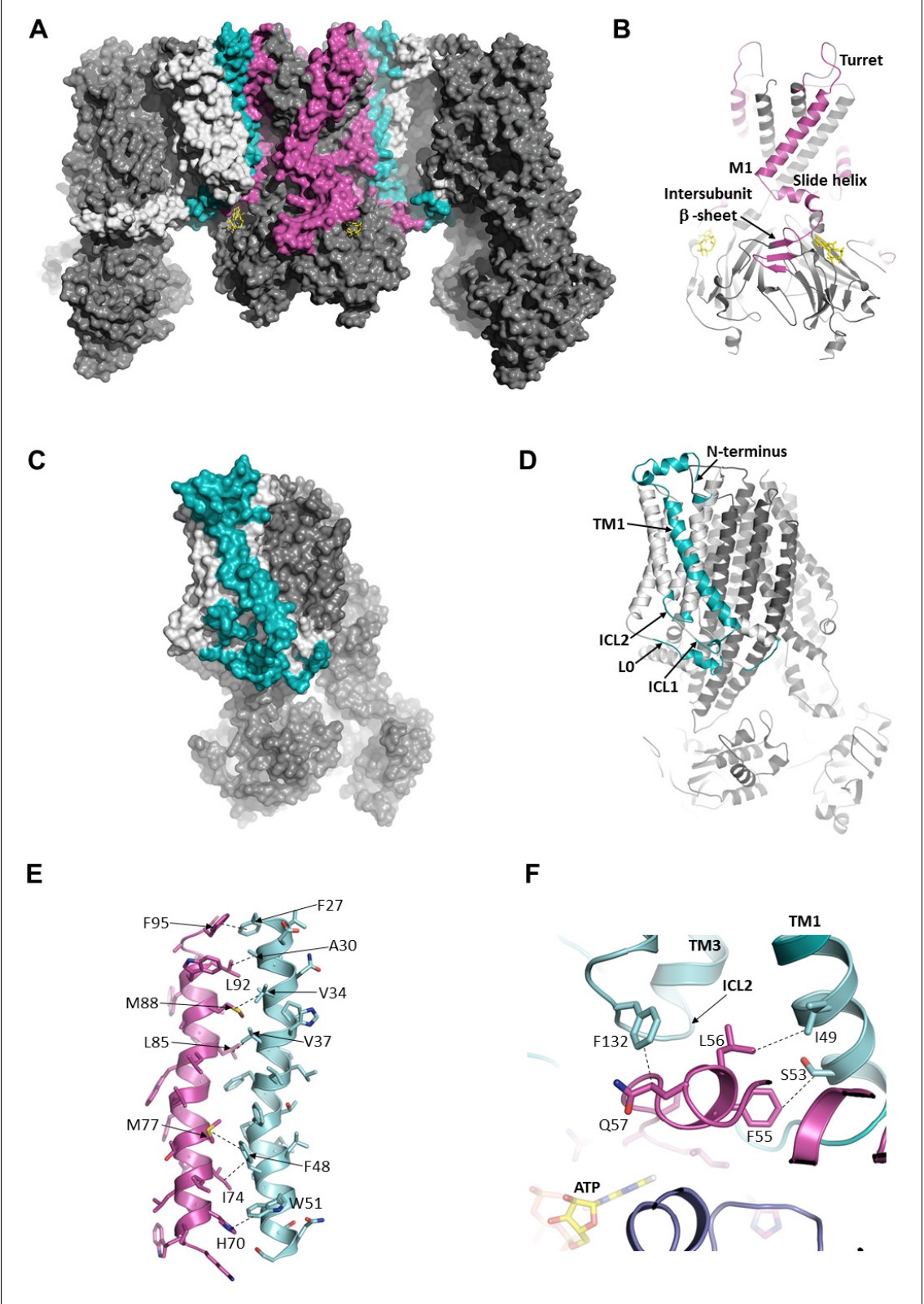

**Figure 3.** The interface between SUR1 and Kir6.2. (**A**) Surface representation of the complex. SUR1-binding surface on Kir6.2 colored in magenta, and Kir6.2-binding surface on SUR1 is in cyan. TMD0/L0 is colored in light gray, and Kir6.2 and the ABC core of SUR1 are in dark gray. (**B**) Cartoon model of Kir6.2, with interface residues colored in magenta. The intersubunit β-sheet is formed by β strands A, N, and O shown in *Figure 2—figure supplement 2*. (**C and D**) Surface and cartoon models of SUR1, with interface residues in cyan. (**E**) Interface between M1 (Kir6.2; magenta) and TM1 (SUR1; cyan),

*Figure 3 continued on next page*

*Figure 3 continued*

highlighting key interactions. (**F**) Intersection of ICL2 (cyan) and N-terminus/slide helix (magenta), showing interaction between Q57 (Kir6.2) and F132 (SUR1). The dashed lines indicate selected van der Waals or electrostatic (H-bonding or charge-charge) interactions between two residues to aid visualization.

DOI: https://doi.org/10.7554/eLife.31054.009

(*Figure 4C*, *Figure 4—figure supplement 1C*). The ribose group is in close contact with the I182 and F333 side chains; the adenine ring stacks against the aliphatic portion of the R50 side chain as well as Y330, and is H-bonded to the main chain nitrogen of R50, and main chain oxygen of N48 and Y330 (*Figure 4C*, *Figure 4—figure supplement 1D and E*). The aforementioned residues have all been shown previously to reduce ATP inhibition when mutated to other amino acids (*Antcliff et al., 2005*; *Cukras et al., 2002*; *Drain et al., 1998*; *Li et al., 2005*; *Proks et al., 1999*; *Tammaro et al., 2005*; *Tucker et al., 1998*), consistent with a role of these residues in ATP gating. Notably, sequence comparison reveals that a key difference between Kir6.2 and other Kir channels is G334, which in other Kir channels is occupied by larger amino acids. Substitution of glycine at this position by a larger amino acids such as histidine seen in Kir2 or Kir3 channels would create steric hindrance to prevent ATP binding. This may explain, at least in part, why Kir6.2 is the only Kir channel sensitive to ATP regulation.

## Structural interactions around the ATP binding site and their relationship to the PIP$_2$ binding site

A number of residues within the vicinity of the ATP-binding site such as E179, R201, and R301 have previously been shown to reduce ATP sensitivity (*Haider et al., 2005*; *Shyng et al., 2000*) and could be involved in ATP binding. However, from the structure it is clear these residues contribute indirectly (*Figure 5*). E179 and R301 were both proposed to interact with the adenine ring (*Haider et al., 2005*). In our structure, neither residue forms direct interactions with ATP (*Figure 5A and C*). E179 appears to interact with R54 from the adjacent Kir6.2 and may be part of the network that stabilizes the interaction between R50 and ATP (*Figure 5C*). R301 is found to interact with Q299 in the same β-strand that is part of a β-sheet in the Kir6.2 CTD (*Figure 5A*; see also *Figure 2—*

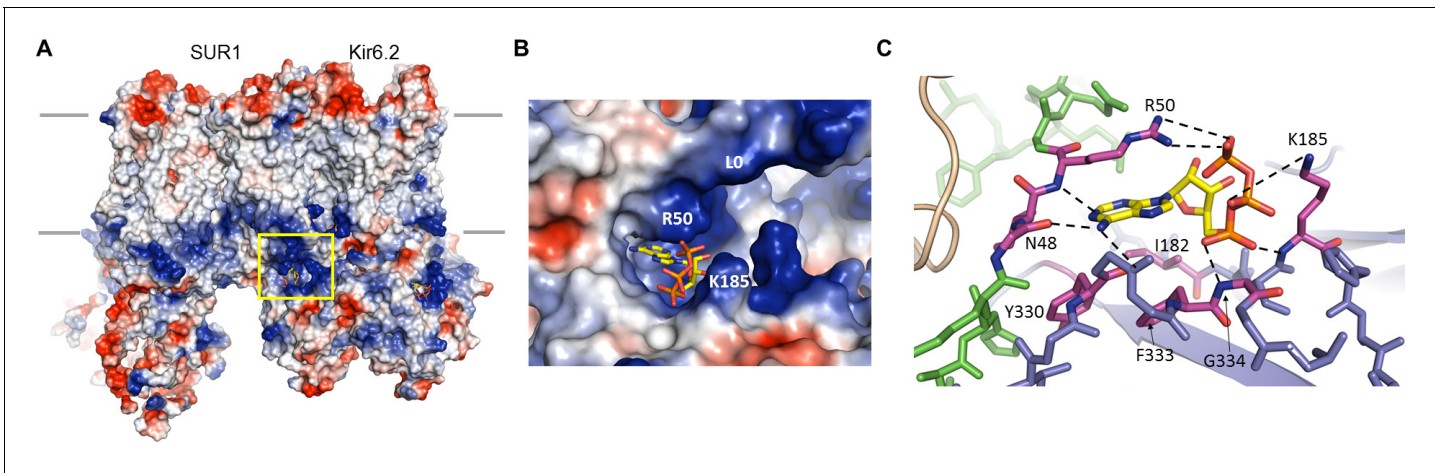

**Figure 4.** The ATP binding pocket. (**A**) Surface representation of a Kir6.2 tetramer in complex with one SUR1, colored by Coulombic surface potential. ATP pocket is boxed in yellow. (**B**) Close-up of ATP binding pocket boxed in (**A**). Note close proximity of L0 to the pocket on Kir6.2. (**C**) Interactions within ATP-binding pocket, with residues directly interacting with ATP colored in magenta. The Kir6.2 subunit containing R50 is colored green, with the adjacent subunit colored blue. The dashed lines indicate possible van der Waals or electrostatic interactions to aid visualization.

DOI: https://doi.org/10.7554/eLife.31054.010

The following figure supplement is available for figure 4:

**Figure supplement 1.** Cryo-EM density of ATP and surrounding residues.

DOI: https://doi.org/10.7554/eLife.31054.011

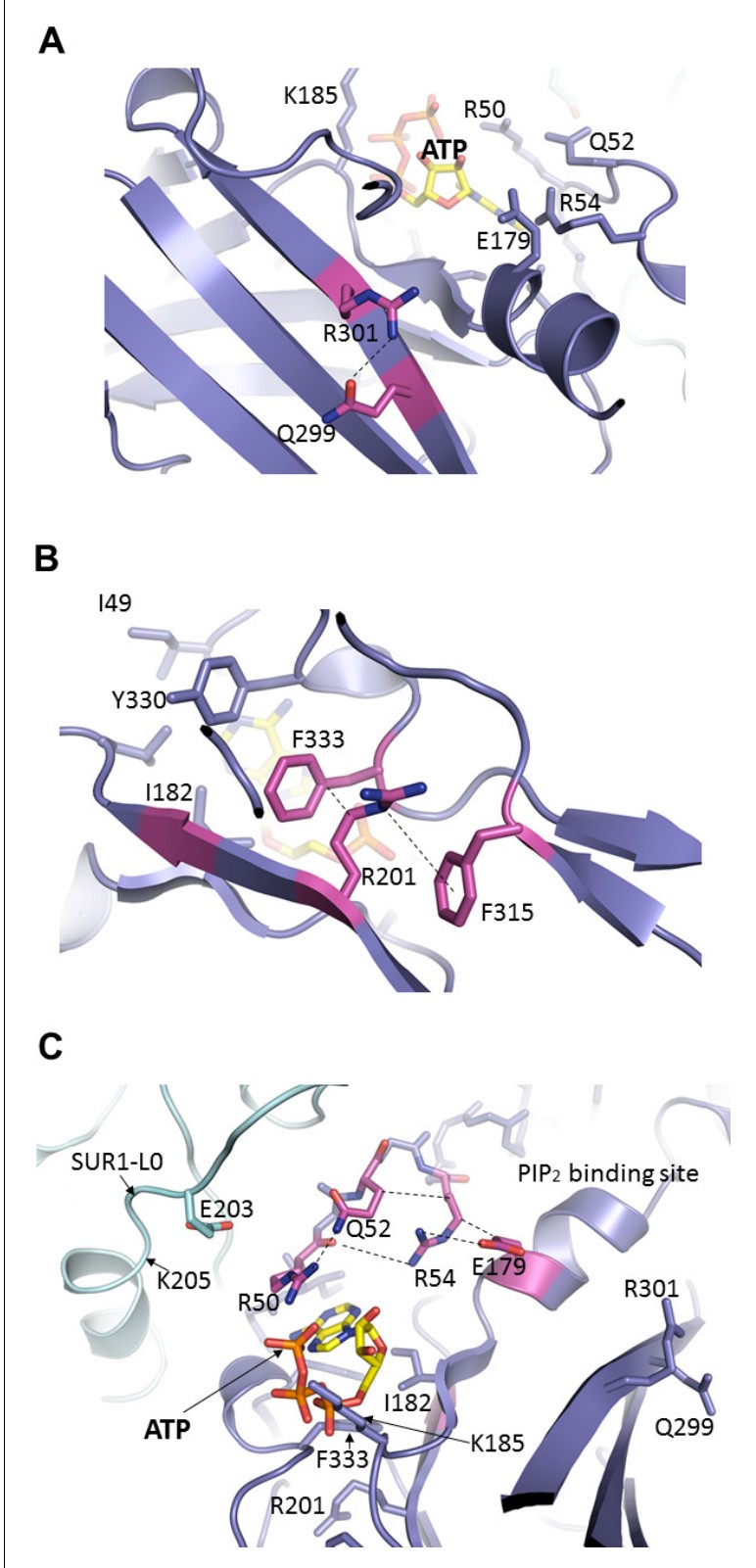

**Figure 5.** Important molecular interactions surrounding the ATP binding site. (**A**) Electrostatic interaction between R301 and Q299, viewed from the interior of Kir6.2 and looking out toward the cytoplasm. These residues are found on an internal β-sheet 12 Å from ATP (Cα of R301 to ribose of ATP) (**B**) Dual interaction between R201 and F315, likely via cation-π, and hydrophobic stacking between aliphatic portion of R201 side chain and F333, again viewed from the Kir6.2 interior. These residues are found directly below ATP (~9 Å from Cα of R201 to ribose of ATP). (**C**) Relationship between R50,

*Figure 5 continued on next page*

*Figure 5 continued*

Q52, R54, and E179 of Kir6.2 near the ATP and PIP$_2$ binding sites. The dashed lines indicate possible van der Waals or electrostatic interactions to aid visualization.

DOI: https://doi.org/10.7554/eLife.31054.012

*figure supplement 1*). Interestingly, R301 is one of the most highly mutated residues in congenital hyperinsulinism (*Snider et al., 2013*); mutation of R301, in addition to mildly reducing ATP sensitivity, results in rapid decay of channel activity that can be reversed by increasing PIP$_2$ concentrations in the membrane (*Lin et al., 2008*; *Shyng et al., 2000*). Based on our structure, it is likely that mutation of this residue disrupts structural integrity of the Kir6.2 CTD necessary for stable channel interaction with PIP$_2$ and ATP. By contrast, R201 is one of the most highly mutated residues in neonatal diabetes (*Ashcroft, 2005*). It has been proposed that R201 coordinates the α-phosphate of ATP (*Haider et al., 2005*). However, in the structure R201 is found on the β-strand directly below that of I182 and K185, and is too distant to directly interact with ATP. Instead, R201 is sandwiched between the benzene rings of F333 and F315, forming a dual cation-π interaction that likely stabilizes the ATP-binding site (*Figure 5B*). Mutation of R201 would therefore destabilize this interaction to indirectly reduce ATP inhibition.

Another interesting residue is Q52. The PNDM mutation Q52R causes extremely high $P_o$ and very low ATP sensitivity (*Koster et al., 2005b*; *Lin et al., 2006*; *Proks et al., 2004*). In the structure, Q52 interacts with R50 which coordinates the γ-phosphate of ATP, but is also interacting with R54, which orients the R54 side chain towards the ATP site and away from the PIP$_2$ site nearby (*Figure 5C*). In the PIP$_2$ bound Kir2.2 structure, the Kir6.2 R54 equivalent arginine residue interacts with the tether helix near PIP$_2$ binding residues (*Hansen et al., 2011*), suggesting that R54 may be important for Kir6.2 -PIP$_2$ interactions. Interestingly, mutation of R50 or R54 to an alanine has been reported to reduce sensitivity to both ATP and PIP$_2$ (*Cukras et al., 2002*). From our structure, it is easy to envision how mutation of any of these residues can disrupt the interaction network to affect gating by either ligand.

It is also important to note that in our structure, Q52 is in close proximity to E203 in the L0 of SUR1 immediately following TMD0. We have previously shown that engineered interactions between Kir6.2 residue 52 and SUR1 residue 203 via a Kir6.2-Q52E and SUR1-E203K ion pair increases channel sensitivity to ATP by nearly two orders of magnitude, and that crosslinking of the two residues via a Kir6.2-Q52C and SUR1-E203C mutant pair induces spontaneous channel closure in the absence of ATP (*Pratt et al., 2012*). In addition, our previous studies have shown that ATP binding involves residues from not only the N-terminus of Kir6.2 such as R50 but also residues in SUR1-L0 such as K205 (*Martin et al., 2017*; *Pratt et al., 2012*). Together these studies lead us to propose that the inhibitory effect of ATP is partially due to stabilizing the interaction between this N-terminal region of Kir6.2 and L0 of SUR1 (see Discussion).

## The GBC binding site

Sulfonylureas stimulate insulin secretion by inhibiting pancreatic K$_{ATP}$ channels (*Aguilar-Bryan and Bryan, 1999*). GBC, also known as glyburide, is a second generation sulfonylurea that contains both a sulfonylurea moiety and a benzamido moiety, and binds K$_{ATP}$ channels with nM affinity (*Gribble and Reimann, 2003*). Despite intense investigation the GBC binding site has remained elusive. Early studies using chimeras of SUR1 and SUR2A, which is known to have lower sensitivity to GBC than SUR1, suggest the involvement of TMs 14–16; in particular mutating S1238 in SUR1 to Y (note in some papers, this is numbered as S1237) as seen in SUR2A compromised GBC binding and inhibition (*Ashfield et al., 1999*; *Winkler et al., 2007*). Subsequent studies using $^{125}$I-azido-GBC photolabeling implicated involvement of L0 of SUR1; specifically, two mutations Y230A and W232A in L0 severely compromised photolabeling of SUR1 (*Vila-Carriles et al., 2007*). These studies led to a model in which S1238 and Y230/W232 constitute two ends of a bipartite binding pocket, each recognizing opposite ends of GBC (*Bryan et al., 2004*; *Winkler et al., 2007*); however, whether one or both contribute directly to GBC binding remained unknown.

In the current reconstruction we find well defined, non-protein density within the TMDs of SUR1, with a size and shape which closely matches that of a GBC molecule (*Figure 6*, *Figure 6—figure*

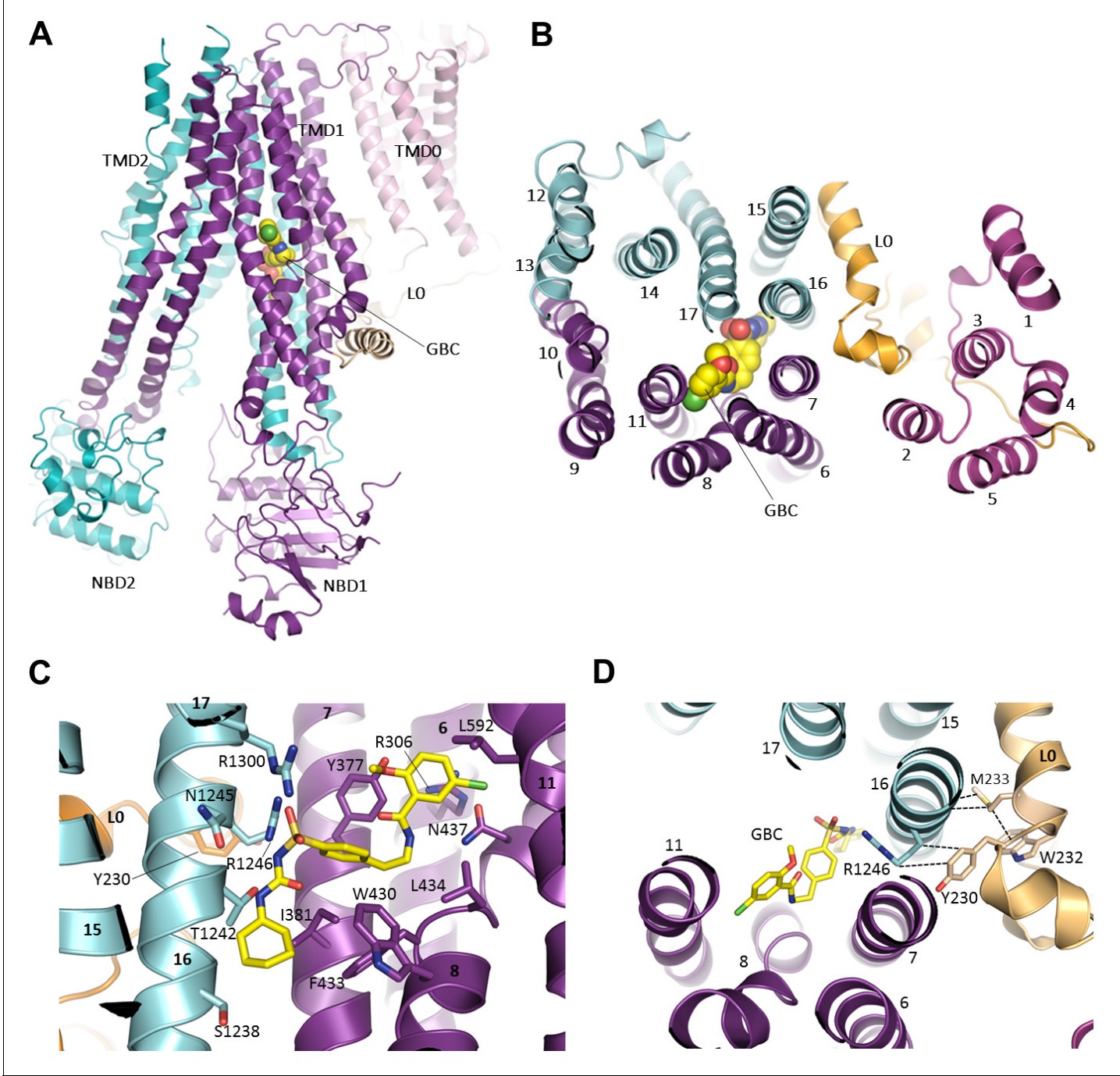

**Figure 6.** The GBC binding site in SUR1. (**A**) Ribbon diagram of SUR1 showing location of GBC, which is primarily coordinated by the inner helices of TMD1 (purple) and TMD2 (cyan). (**B**) Slice view of model in (**A**) viewed from the extracellular side. Note juxtaposition of L0 to helices in ABC core directly interacting with GBC. (**C**) Close-up of GBC binding pocket, showing all residues which immediately line the pocket and seem to form direct contact with GBC; a subset of these residues were mutated to test their role in GBC binding (**Figure 7**). (**D**) Magnified view in (**B**), highlighting indirect roles of Y230 and W232 (L0) in GBC binding. These both likely stabilize interactions between residues on helix 16 of TMD2 and GBC, at the same time anchoring this helix of L0 to the ABC core structure.

DOI: https://doi.org/10.7554/eLife.31054.013

The following figure supplement is available for figure 6:

**Figure supplement 1.** GBC binding site.
DOI: https://doi.org/10.7554/eLife.31054.014

*supplement 1A*). One end of the density is in direct contact with S1238 and resembles the cyclohexyl moiety long presumed to constitute the 'A' site that is abolished by the S1238Y mutation (*Ashfield et al., 1999*; *Bryan et al., 2004*). We used this to guide the initial docking of GBC, which could then be readily refined into the density together with SUR1.

The binding pocket is contoured to precisely accommodate GBC, and the combination of polar and hydrophobic residues help explain the sub-nM affinity of SUR1 for this sulfonylurea (*Figure 6C*, *Figure 6—figure supplement 1B,C*). A primary anchor is composed of two arginine residues, R1246 and R1300, which coordinate each oxygen of the sulfonyl group. Each nitrogen of the urea moiety is coordinated by T1242 and N1245, and the adjacent benzene and cyclohexyl groups (adjacent to the sulfonyl and urea groups, respectively) are stabilized by a series of hydrophobic interactions contributed by TM helices from both TMD1 (TM6, 7, 8) and TMD2 (TM16). As a second-generation sulfonylurea, GBC contains another lipophilic group adjacent to an amide linker, which is lacking in first-generation compounds like tolbutamide (*Gribble and Reimann, 2003*). This group, a 1-chloro-4-methoxy-benzene, is encircled by a ring of hydrophilic and hydrophobic side chains. In particular, the Cl appears to hydrogen bond with the amino group of N437, while the methoxy is H-bonded to the hydroxyl group of Y377. Y377 also seems to contribute a π-π stacking interaction with the benzene ring. In the structure, the previously proposed sulfonylurea binding residue S1238 juxtaposes the cyclohexyl group of GBC, with only ~3 Å separation between the Cβ of S1238 and the 6-carbon ring of GBC (*Figure 6C*; *Figure 6—figure supplement 1D*). Mutation of this residue to a tyrosine may alter interaction with GBC to compromise high affinity binding of GBC.

In order to validate the proposed binding site, we mutated a subset of key GBC-binding residues listed above to alanine and tested their response to 100 nM and 1 μM GBC with $Rb^+$ efflux experiments, which measure channel activity and response to GBC in intact cells. All six mutations, R306A, Y377A, N437A, T1242A, R1246A, and R1300A trafficked normally and responded as WT to metabolic inhibition (*Figure 7A, B and D*). Strikingly, all six mutants showed significantly reduced or complete absence of inhibition at 100 nM, and five mutants (R306A, Y377, N437A, T1242, and R1246A) showed significantly reduced sensitivity compared to WT even at 1 μM GBC (*Figure 7C,D*). The four most GBC-insensitive mutants, R306A, Y377A, N437A, and T1242A were further analyzed by inside-out patch-clamp recording. Although these mutants were still sensitive to GBC inhibition, the extent of inhibition at steady-state was less compared to WT channels at 10 nM and 100 nM (*Figure 7—figure supplement 1*). Also worth noting, while inhibition of WT channels was nearly irreversible, inhibition of mutants was more reversible, consistent with the mutants having reduced affinity for GBC (*Figure 7—figure supplement 1*). Together these results provide strong functional evidence for the GBC binding pocket defined in our structure.

## The role of SUR1-Y230 and W232 in GBC interaction

In the previously proposed bipartite binding model for GBC (*Bryan et al., 2004*), the pocket was formed from two overlapping regions: at one end was S1238 of TMD2, and at the other was L0 involving residues Y230 and W232, part of the 'lasso motif' observed in CFTR (*Zhang and Chen, 2016*) and MRP1 (*Johnson and Chen, 2017*). Mutation of Y230 to an alanine has also been shown to reduce the ability of GBC to inhibit channel activity (*Devaraneni et al., 2015*; *Yan et al., 2006*). In our current structure, Y230 is too distant to interact directly with GBC. However, the binding pocket is close to the L0-TMD interface, where the L0 amphipathic helix forms a series of mostly hydrophobic interactions with transmembrane helices from TMD1/2 that line the GBC binding pocket. Here, we find that Y230 stacks closely against the aliphatic portion of the R1246 side chain, which in turn coordinates an oxygen of the sulfonyl group of GBC (*Figure 6D*; *Figure 6—figure supplement 1E*). W232 appears to form a strong interaction with M233, which interacts directly with two alanines, A1243 and A1244, on the opposite side of TM16 where two GBC interacting residues T1242 and N1245 are located (*Figure 6D*; *Figure 6—figure supplement 1D*). These observations indicate an important but clearly indirect role for Y230 and W232 in GBC binding.

## Comparison with previous $K_{ATP}$ channel structures

To date, two $K_{ATP}$ channel structures have been reported, one from our group (*Martin et al., 2017*) in the presence of GBC and ATP at 5.7 Å resolution, and the other by Li et al. (*Li et al., 2017*) in the presence of GBC but absence of ATP at 5.6 Å resolution. The structure presented here is also in the

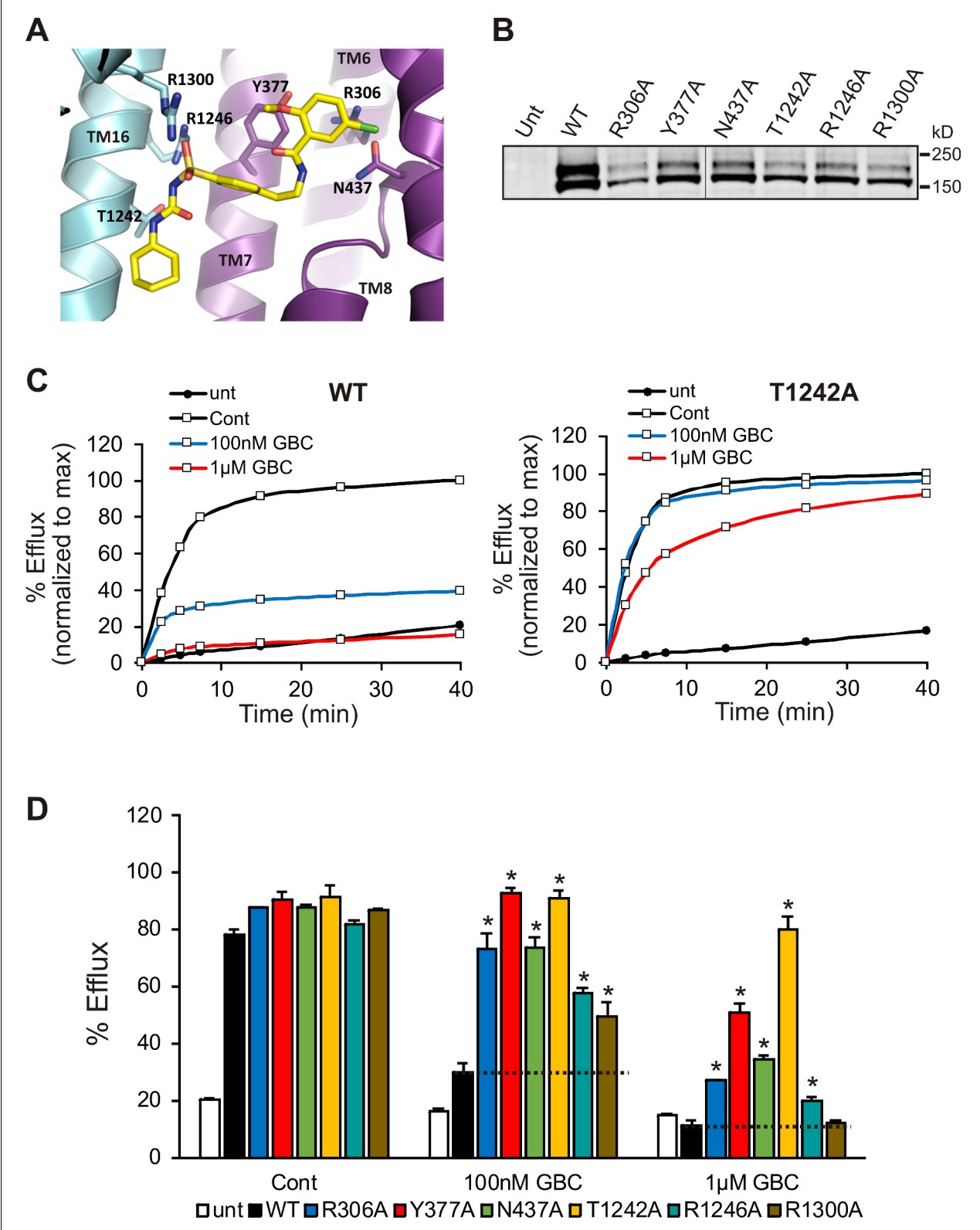

**Figure 7.** Functional testing of GBC binding residues. (**A**) Residues in SUR1 selected to be mutated to alanine. (**B**) Western blot of WT and mutant SUR1 co-expressed with Kir6.2 in COS cells. Two SUR1 bands corresponding to the core-glycosylated immature protein (lower band) and the complex-glycosylated mature protein (upper band) are detected. The vertical line in the middle of the blot separates two parts of the same blot. (**C**) Representative efflux profiles of WT channels and T1242A mutant channels in cells pretreated with metabolic inhibitors for 30 min in the presence of

*Figure 7 continued on next page*

*Figure 7 continued*

0.1% DMSO (cont), 100 nM GBC, or 1 µM GBC. Untransfected cells (unt) served as a control. Efflux was normalized to the maximal value observed at 40 min for direct comparison. (D) Quantification of percent efflux of all mutants compared to WT. Each bar represents the mean ± s.e.m. of 3–4 biological repeats. *p<0.05 by one-way ANOVA with Newman-Keuls *post hoc* test.

DOI: https://doi.org/10.7554/eLife.31054.015

The following figure supplement is available for figure 7:

**Figure supplement 1.** Functional testing of GBC binding residues by electrophysiology.

DOI: https://doi.org/10.7554/eLife.31054.016

presence of ATP and GBC and is nearly identical to the structure we published previously but with much improved resolution, allowing for accurate modelling of nearly all side-chains, many of which were absent in our previous structure (*Martin et al., 2017*). To gain insight into the conformational difference between ATP-bound and ATP-free channels and how resolution of the map may affect structural interpretation, we compared our current structure with that of Li et al. in detail.

Overall, we find that the two structures are also very similar, both in terms of organization of the complex and conformations of Kir6.2 and SUR1 individually (*Figure 8*). Since the structure from Li et al. lacks ATP whereas our structure is bound to ATP, it suggests that either ATP does not induce significant conformational change of the channel or that GBC, which is present in both structures, stabilizes the channel in a conformation that resembles an ATP-bound state. The latter possibility is intriguing as it offers a potential mechanism by which GBC inhibits channel activity.

Despite overall similarity, there are some key differences between our current structure and that of Li *et al.* which warrant addressing, especially with regard to interpretation of the cryo-EM density and structural modeling of the GBC-binding site. In Li *et al*, they attributed GBC to unassigned density surrounding Y230 and W232, two residues previously proposed to be involved in sulfonylurea binding (*Vila-Carriles et al., 2007*). However, in their structure, they did not model residues 214–222 of SUR1-L0, which happen to lie close to their 'GBC' density (*Figure 8—figure supplement 1A and B*). Docking our current structure into their map (*Figure 8—figure supplement 1A and B*, blue), we find that the density they observed matches well to those residues left out of their model, corresponding to approximately R216-F221. Moreover, the size and shape of the density surrounding those residues in our current reconstruction are reminiscent of their 'GBC' density, albeit a higher-resolution version (*Figure 8—figure supplement 1C and D*). Interestingly, having identified the GBC binding site using our higher resolution structure, we re-inspected our previous 5.7 Å map as well as the map by Li *et al.* and found unassigned density near the GBC binding site that corresponds to the size and shape of GBC in both (*Figure 8—figure supplement 2*).

Another difference between our structures and the structures reported by Li *et al.* concerns the location of $PIP_2$ (*Li et al., 2017*; *Martin et al., 2017*). In both studies, membranes containing channel proteins were solubilized in digitonin with no addition of exogenous $PIP_2$ in subsequent purification steps. It is possible that some endogenous $PIP_2$ might have been co-purified with the channel in our studies. We have observed what appears to be a heterogeneous mixture of lipids and detergent near the predicted $PIP_2$ binding site in all of our reconstructions, but were unable to clearly distinguish the identity of this density, even at improved resolution. We therefore did not model $PIP_2$ in our earlier structure or the current structure. By contrast, Li *et al.* tentatively assigned extra density between two Kir6.2 subunits in one of their 3D classes as $PIP_2$ and proposed that it underlies the more dilated inner helices of two of the four Kir6.2 subunits in that class (*Li et al., 2017*). Unfortunately, the resolution of the 3D class in which $PIP_2$ was observed was 8.5 Å and no cryo-EM density map was available, making it difficult to compare with our structures directly. Future studies will be needed to resolve this problem.

## Discussion

The structure presented in this study is the first to reveal in detail the ATP and GBC binding sites in the SUR1/Kir6.2 $K_{ATP}$ channel complex. The clear density for ATP and GBC as well as all residues involved in binding of both ligands in the current EM map allowed us to present a detailed atomic interpretation of ATP and GBC binding to the channel. Importantly, the binding pockets we identified are supported by strong functional data. In addition, the structure uncovers many molecular

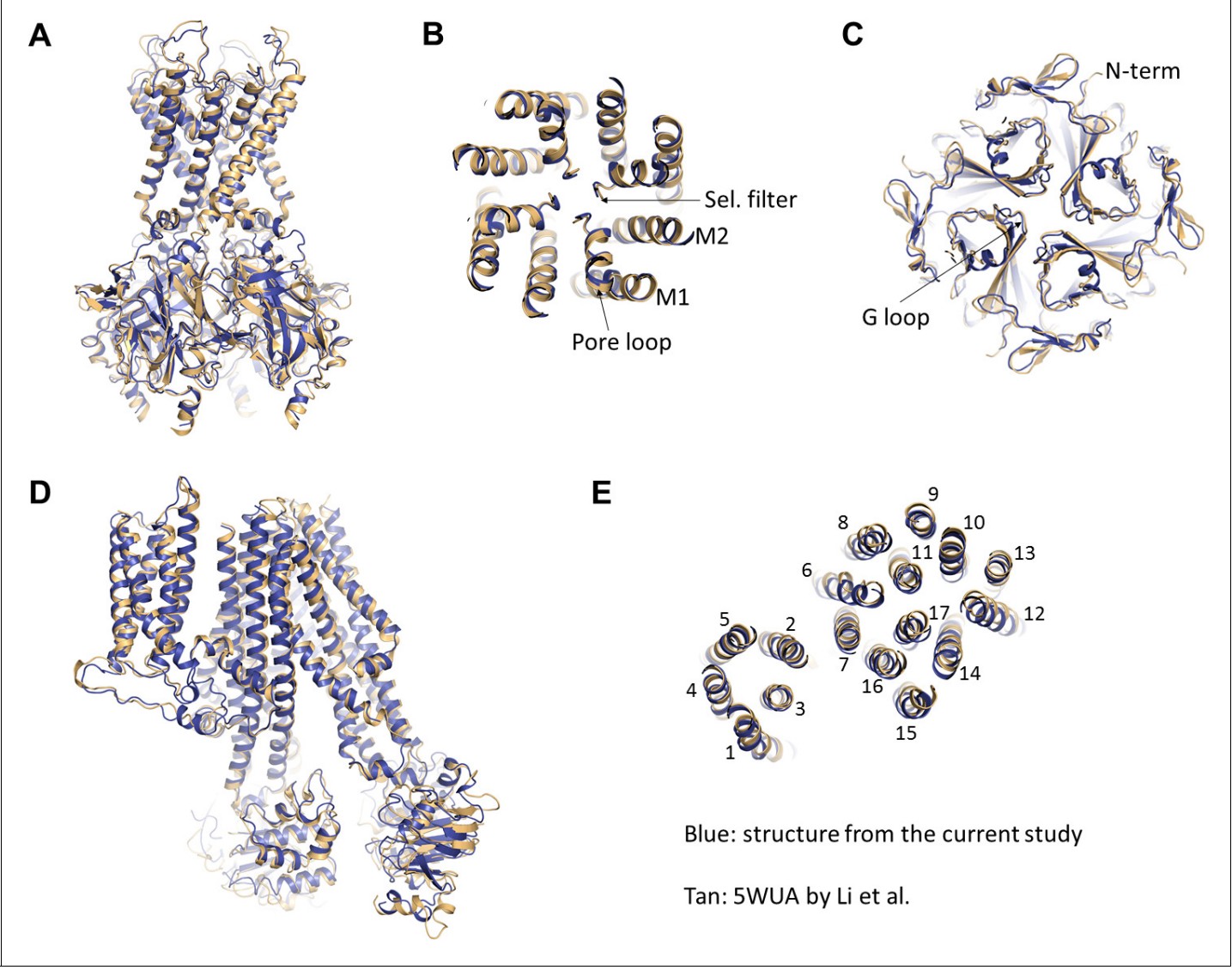

**Figure 8.** Comparison of the current structure with the GBC-bound, ATP-free structure from Li *et al.*(PDB ID: 5WUA). (**A**) Overlay of the Kir6.2 structure viewed on the side. (**B**) Overlay of the Kir6.2 membrane helices viewed from the top. (**C**) Overlay of the Kir6.2 cytoplasmic domain. (**D**) Side view of the overlay of the SUR1 structure. (**E**) Overlay of the SUR1 transmembrane helices 1–17 viewed from the top. In all panels, the higher resolution structure from the current study is colored in blue, and the 5WUA structure from Li *et al.* is colored in tan.

DOI: https://doi.org/10.7554/eLife.31054.017

The following figure supplements are available for figure 8:

**Figure supplement 1.** Reinterpretation of the GBC cryo-EM density proposed in Li *et al.*

DOI: https://doi.org/10.7554/eLife.31054.018

**Figure supplement 2.** Cryo-EM density that likely corresponds to GBC in previously published $K_{ATP}$ channel structures.

DOI: https://doi.org/10.7554/eLife.31054.019

---

interactions that indirectly impact ATP and GBC gating, and those that underlie SUR1-Kir6.2 interactions. The structural information gained offers key insights into possible mechanisms of how the two ligands both inhibit $K_{ATP}$ channels to stimulate insulin secretion.

## The ATP binding site and mechanism of ATP inhibition

The Kir6.2 interfacial ATP binding site model was first proposed by Antcliff *et al.* (*Antcliff et al., 2005*) based on ligand docking, homology modeling of Kir channel crystal structures, and structure-function mutagenesis data. Although some interactions in the original model between Kir6.2

residues and ATP are observed in our structure, many others require new interpretations. First, the ATP molecule adopts a conformation with the γ-phosphate bent towards the adenine ring (*Figure 4*), which is reminiscent of that observed in P2X receptors, an ATP activated ion channel (*Hattori and Gouaux, 2012*). Second, our structure suggests that E179, R201, and R301 rather than contributing directly to ATP binding are critical for interactions with other residues that support the ATP binding residues or general structural integrity of the Kir6.2 CTD for stable channel interaction with $PIP_2$ (*Figure 5*). Elucidation of the structural role of R201 and R301 helps us to understand the mechanisms by which mutation of these residues cause insulin secretion disease. Finally, although in our structure $PIP_2$ is not present or resolved, the residues previously proposed to be involved in $PIP_2$ binding or gating based on functional studies (*Cukras et al., 2002*; *Shyng et al., 2000*) and crystal structures of $PIP_2$ bound Kir2.2 and 3.2 channels (*Hansen et al., 2011*; *Whorton and MacKinnon, 2011*) can be clearly modeled, which reveals the intricate relationship between ATP binding residues and those involved in $PIP_2$ binding or gating (*Figure 5C*) and offers insight into how the channel senses ATP and $PIP_2$.

We propose that ATP, by binding to a pocket created by the N-terminus and CTD from two adjacent Kir6.2 subunits with contributions from L0 of SUR1, acts to stabilize interactions between the Kir6.2 N-terminus and L0 to prevent movements necessary to open the channel (*Figure 9A*). This model is consistent with our previous study showing that crosslinking of SUR1-L0 with the N-terminus of Kir6.2 near the ATP binding site locks the channel closed even without ATP (*Pratt et al., 2012*). Previous studies have shown that ATP and $PIP_2$ functionally antagonize each other through allosteric regulation (*Enkvetchakul et al., 2000*) and that ATP can bind both closed and open channels (*Enkvetchakul et al., 2001*; *Li et al., 2000*). One interesting question is whether the interaction network we observed in the present ATP-bound structure undergoes remodeling in $PIP_2$-bound open state. An open state structure of the $K_{ATP}$ channel bound to $PIP_2$ will be needed to understand the full extent of conformational change associated with ATP and $PIP_2$ gating.

## Mechanistic insights of GBC binding and inhibition

The binding site of the high affinity sulfonylurea GBC has been studied by many groups. These studies have implicated the involvement of transmembrane helices in the SUR1-ABC core, L0, and the N-terminus of Kir6.2 (*Ashfield et al., 1999*; *Bryan et al., 2004*; *Vila-Carriles et al., 2007*). Yet, the precise binding pocket for this commonly used anti-diabetic drug has remained unresolved. In the structure presented here, we were able to clearly assign the GBC density in the TM bundle connected to NBD1, with residues from TM6, 7, 8, 11 in TMD1 and TM16 and 17 from TMD2 contributing to GBC interactions. Importantly, our model is supported by functional data using both $^{86}Rb^+$ efflux assays and electrophysiological recordings. Moreover, our structure clarifies how Y230, which has previously been proposed to contribute to GBC binding based on indirect biochemical or functional assays (*Vila-Carriles et al., 2007*), can affect GBC binding or gating indirectly by supporting residues that are directly engaged in GBC binding.

The mechanism by which GBC inhibits channel activity is complex. In *Figure 9B*, we present a hypothetical model to explain the current structural and functional data. In the presence of MgATP/ADP, there is evidence that the NBDs of SUR1 undergo dimerization to switch the SUR1-ABC core structure from an inward-facing conformation to an outward-facing conformation to antagonize the inhibitory effect of ATP at the Kir6.2 site, and GBC binding to SUR1 stabilizes the SUR1-ABC core in an inward-facing conformation to prevent MgATP/ADP from opening the channel (*Ortiz et al., 2012*) (*Figure 9B*). In the absence of MgATP/ADP where the SUR1-ABC core is expected to be in an inward-facing conformation, channels are still able to open with high probability (*Lin et al., 2003*) and GBC also causes rapid inhibition of channel activity under such a condition (*Figure 7—figure supplement 1A*), suggesting GBC can inhibit channels in a MgATP/ADP independent manner. While the mechanism by which GBC inhibits channel activity in the absence of MgATP/ADP is not clear, we hypothesize that it may involve modulating interactions between the distal N-terminus of Kir6.2 and SUR1. The distal N-terminal 30 amino acids of Kir6.2 have been shown to be important for the binding or effect of GBC in a number of studies (*Devaraneni et al., 2015*; *Koster et al., 1999*; *Kühner et al., 2012*; *Reimann et al., 1999*; *Vila-Carriles et al., 2007*). Moreover, it is known to be involved in regulating channel open probability by interacting with L0 of SUR1 (*Babenko et al., 1999*; *Shyng et al., 1997*). In our map, there is a lack of strong density N-terminal to position 32 of Kir6.2, suggesting this region is flexible. However, it is worth noting that our previous study using

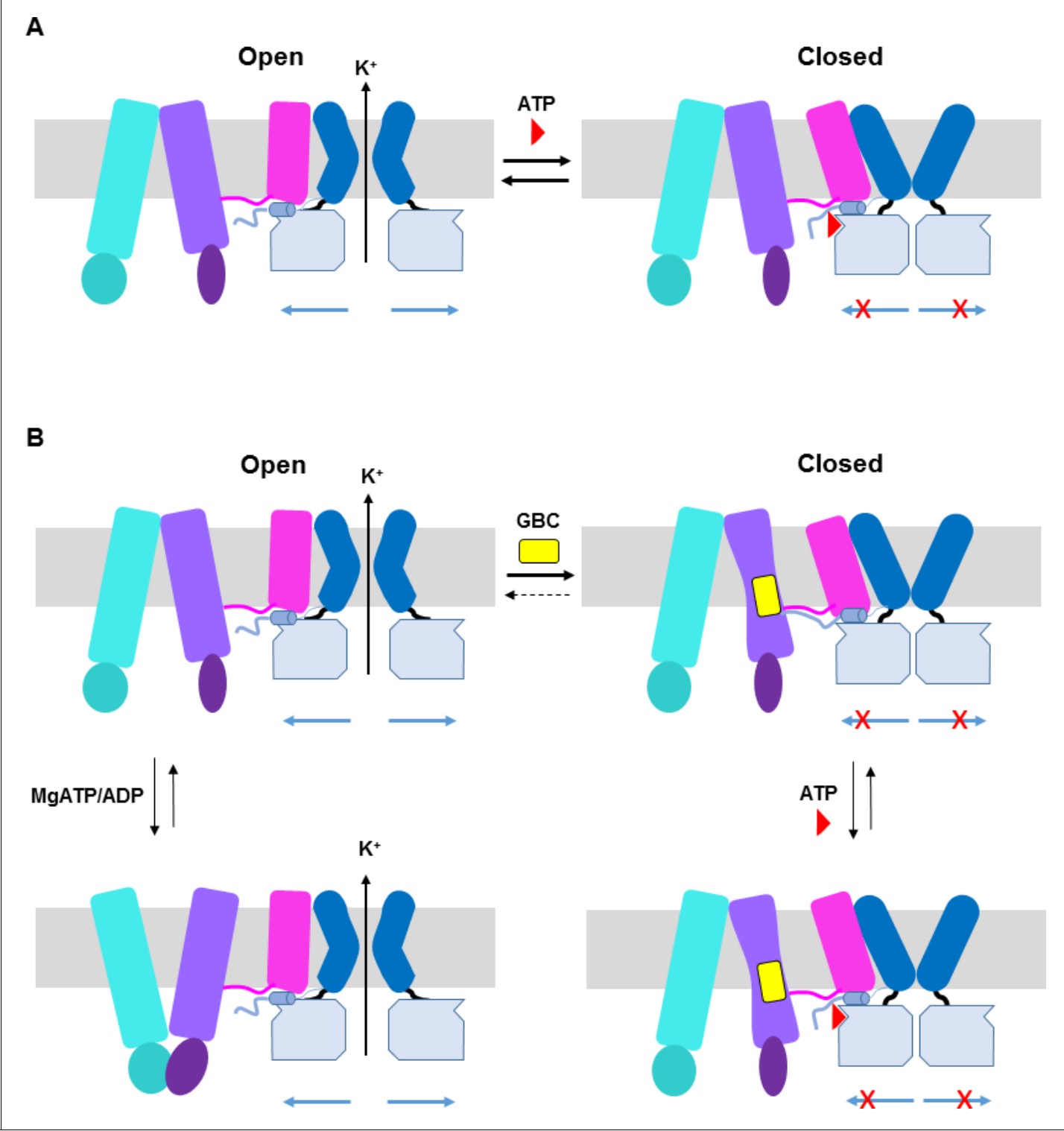

**Figure 9.** ATP and GBC gating models. (**A**) Hypothetical model illustrating that ATP binds to a pocket formed by the N-terminus and CTD of Kir6.2 (from two adjacent subunits), with contribution form L0 of SUR1. This stabilizes the channel in a closed state that is energetically unfavorable for transitioning into an open state. (**B**) Hypothetical model of GBC gating. In the absence of nucleotides, GBC binds to the TM bundle juxtaposing L0, which stabilizes the distal N-terminus of Kir6.2 to greatly reduce channel open probability and promote channel closure. Addition of ATP further closes the channel by preventing residual free N-terminus from moving channels into an open state (see panel A). In the presence of MgATP/ADP, the SUR1-ABC core can transition from an inward-facing conformation to an outward conformation upon dimerization of the NBDs to antagonize ATP inhibition on Kir6.2 and promote channel opening; GBC binding stabilizes the SUR1-ABC core in the inward-facing conformation and shifts the equilibrium

*Figure 9 continued on next page*

*Figure 9 continued*
towards channel closure. The dashed lines between states illustrate the near irreversible binding of GBC. In both A and B, Kir6.2 transmembrane helices: dark blue; Kir6.2 cytoplasmic domain: pale blue; Kir6.2 slide helix and N-terminus from adjacent subunit: light blue cylinder and thick light blue line, respectively; SUR1-TMD0/L0: magenta; SUR1-TMD1: light purple; SUR1-NBD1: dark purple; SUR1-TMD2: cerulean; SUR1-NBD2: deep cerulean; GBC: yellow; ATP: red. Note the different states shown are not meant to reflect the actual kinetic transitions, but the hypothesized stable states.
DOI: https://doi.org/10.7554/eLife.31054.020

engineered unnatural amino acid Azido-*p*-phenylalanine placed at the distal N-terminus of Kir6.2 (amino acid position 12 or 18) has demonstrated that GBC increased crosslinking of Kir6.2 to SUR1 (*Devaraneni et al., 2015*). Thus, we hypothesize that GBC binding to the TMD bundle next to the L0 amphipathic helix of SUR1 stabilizes the interactions between the N-terminus of Kir6.2 and SUR1 to prevent the movement of Kir6.2 N-terminus that is needed to open the gate (*Figure 8B*). Although GBC is a potent inhibitor of $K_{ATP}$ channels, it does not completely eliminate channel activity, unlike ATP. Channels exposed to saturating concentrations of GBC can be further inhibited by addition of ATP (see *Figure 7—figure supplement 1A*; *Figure 8B*). Whether GBC and ATP binding events are completely independent remains an open question. Comparison of channel structures without either inhibitors or with only a single inhibitor will be needed to address the issue.

The residues which play a specific and critical role in GBC binding are also very likely important for binding of other sulfonylureas. While we only tested GBC, we predict that R1246 and R1300, which coordinate the sulfonyl group, and T1242 and N1245, which coordinate the urea group, will also be critical for binding of other sulfonylureas such as tolbutamide (*Gribble and Reimann, 2003*). In addition to sulfonylureas, glinides such as rapaglinide and nateglinide which lack the sulfonylurea moiety (*Gribble and Reimann, 2003*), and a structurally unrelated compound carbamazepine (*Chen et al., 2013*; *Devaraneni et al., 2015*) are also known to inhibit $K_{ATP}$ channels. Elucidating the role of the various GBC binding residues in channel interactions with the different channel inhibitors will be important for understanding channel inhibition mechanisms and for rational design of new drugs with desired properties.

## Conservation of GBC binding residues in other SUR proteins and ABCC transporters

Multiple sequence alignment of 15 SUR1 orthologs from diverse genera shows relatively high sequence identity throughout the sequence relative to human; from 95% (hamster) to 75% (seahorse). Interestingly, the segments of the helices from TMD1 and TMD2 which comprise the GBC binding site show exceptionally high conservation, with every one of the 12 residues which most closely line the GBC pocket absolutely conserved in 15 out of the 15 sequences. The high degree of conservation suggests the importance of the interface formed by these transmembrane helices. It would be important to determine in the future whether this interface is involved in the conformational switch of the SUR1-ABC core and its communication with Kir6.2.

Interestingly, SUR2 (*ABCC9*), the closest homolog of SUR1 (67% sequence identity), while also shows high conservation of the GBC binding residues (10/12), differs in two positions that correspond to S1238 and T1242 (Y and S respectively in SUR2). SUR2 assembles with Kir6.1 or Kir6.2 to form $K_{ATP}$ channel subtypes found in the heart, skeletal muscle, and vascular smooth muscle. These channels are known to have lower sensitivity to GBC inhibition than channels formed by SUR1 and Kir6.2 (*Inagaki et al., 1996*). Variations at these two key GBC binding residues likely explain their different pharmacological sensitivity to GBC (*Ashfield et al., 1999*; *Inagaki et al., 1996*).

In addition to targeting $K_{ATP}$ channels, GBC has been shown to inhibit other ABC transporters within the C subfamily (ABCC), including MRP1 and CFTR, albeit at lower affinity (~30 µM for both CTFR and MRP1) (*Payen et al., 2001*; *Schultz et al., 1996*). Of all the residues within the GBC binding pocket, only two appear to be highly conserved across different members of the ABCC subfamily: R1246 and R1300. In fact, R1246 is strictly conserved within 11 of 12 ABCC homologs (Gln in ABCC10), and R1300 in 10 of 12 (Asn in CFTR and Ser in ABCC10). In the cryo-EM structure of MRP1 bound to substrate LTC-4 (*Johnson and Chen, 2017*), R1196 (equivalent to R1246 in SUR1) forms a salt bridge with a carboxylic acid group of LTC-4 and is also in the same rotameric conformation as R1246 in SUR1. Further, F221 (equivalent to Y230 in SUR1) also seems to form the equivalent

hydrophobic stacking interaction with R1196 as Y230 does with R1246 in SUR1; this phenomenon is also observed in the human CFTR structure (F17 and R1097) (*Liu et al., 2017*). Such structural conservation likely explains the GBC sensitivity in other ABCC homologues, and suggests a critical role for this pair of residues in the function and/or structure of ABCC proteins.

In summary, we presented a $K_{ATP}$ channel structure with improved resolution that allowed us to definitively identify the ATP and GBC binding sites. The novel insight gained from this structure significantly advances our understanding of how these two ligands interact with the channel to exert an inhibitory effect. As inhibition of $K_{ATP}$ channels by sulfonylureas remains an important therapeutic intervention to control type 2 diabetes and neonatal diabetes, and there is a need for drugs that specifically target a $K_{ATP}$ channel subtype, our study offers a starting point for future structure-guided drug development to mitigate diseases caused by $K_{ATP}$ channel dysfunction.

# Materials and methods

**Key resources table**

| Reagent type (species) or resource | Designation | Source or reference | Identifiers |
|---|---|---|---|
| Recombinant DNA/adenovirus | | | |
| FLAG-tagged SUR1 (Cricetus cricetus) in AdEasy | ham f-SUR1 | PMID: 28092267 | N/A |
| Kir6.2 (Rattus norvegicus) in AdEasy | rat Kir6.2 | PMID: 28092267 | N/A |
| tTA adenovirus | | PMID: 28092267 | N/A |
| FLAG-tagged SUR1 (Cricetus cricetus) in pECE | f-SUR1 | PMID: 11226335 | N/A |
| Kir6.2 (Rattus norvegicus) in pCDNA3 | | PMID: 14707124 | N/A |
| Cell lines | | | |
| INS-1 clone 832/13 (Rattus norvegicus) | | PMID: 10868964 | RRID:CVCL_7226 |
| COS-M6 (Chlorocebus aethiops) | COSm6 | PMID: 11226335 | RRID:CVCL_8561 |
| Software/Algorithms | | | |
| Serial EM | | PMID: 16182563 | http://bio3d.colorado.edu/SerialEM |
| MOTIONCOR2 | | PMID: 28250466 | http://msg.ucsf.edu/em/software/motioncor2 |
| CTFFIND4 | | PMID: 26278980 | http://grigor><br>ieefflab.janelia.org/ctffind4 |
| DoGPicker | | PMID: 19374019 | https://sbgrid.org/software/titles/dogpicker |
| Relion-2 | | PMID: 27845625 | https://www2.mrc-lmb.cam.ac.uk/relion |
| Frealign | | PMID: 27572728 | http://grigorieefflab.janelia.org/frealign |
| Bsoft | | PMID: 11472087 | https://lsbr.niams.nih.gov/bsoft/ |
| COOT | | PMID: 20383002 | http://www2.mrc-lmb.cam.ac.uk/personal/pemsley/coot |
| RosettaCM | | PMID: 24035711 | https://www.rosettacommons.org |
| UCSF Chimera | | PMID: 15046863 | http://www.cgl.ucsf.edu/chimera |
| Pymol | PyMOL | | https://pymol.org/2 |
| MolProbity | | PMID: 20057044 | http://molprobity.biochem.duke.edu |
| T-Coffee | | PMID: 10964570 | http://www.tcoffee.org/Projects/tcoffee/ |
| Chemicals/Commercial Kits/Antibodies | | | |
| Digitonin | | Calbiochem (San Diego, CA) | CAS 11024-24-1 |
| ATP | | Sigma-Aldrich (St. Louis, MO) | A7699 |
| Glibenclamide | | Sigma-Aldrich | G0639 |
| QuikChange mutagenesis kit | | Agilent (Santa Clara, CA) | 200515 |

*Continued on next page*

*Continued*

| Reagent type (species) or resource | Designation | Source or reference | Identifiers |
|---|---|---|---|
| FuGENE6 | | Promega (Madison, WI) | E2691 |
| Anti-FLAG M2 affinity gel | | Sigma-Aldrich | A2220 |
| FLAG peptide | | Sigma-Aldrich | F3290 |
| Anti-SUR1 (*Oryctolagus cuniculus*) | | PMID: 17575084 | N/A |
| Super Signal West Femto | | Pierce (Rochford, IL) | PI34095 |
| Other | | | |
| R1.2/1.3 300 mesh UltrAuFoil grids | | Quantifoil (Großlöbichau, Germany) | Q27507 |

## Cell lines used for protein expression

INS-1 cells clone 832/13 and COSm6 cells were used for protein expression (see below). The identify of these cell lines has been authenticated (see Key Resources Table above). These cell lines are not on the list of commonly misidentified cell lines maintained by the International Cell Line Authentication Committee. The mycoplasma contamination testing was performed routinely in the lab and shown to be negative for the work described here.

## Protein expression and purification

$K_{ATP}$ channels were expressed and purified as described previously (*Martin et al., 2017*). Briefly, the genes encoding pancreatic $K_{ATP}$ channel subunits, which comprise a hamster SUR1 and a rat Kir6.2 (94.5% and 96.2% sequence identity to human, respectively), were packaged into recombinant adenoviruses (*Lin et al., 2005*; *Pratt et al., 2009*); both are WT sequences, except for a FLAG tag (DYKDDDDK) that had been engineered into the N-terminus of SUR1 for affinity purification. INS-1 cells clone 832/13 (*Hohmeier et al., 2000*), a rat insulinoma cell line, were infected with the adenoviral constructs in 15 cm tissue culture plates. Protein was expressed in the presence of 1 mM Na butyrate and 5 μM GBC to aid expression of the channel complex. 40–48 hr post-infection, cells were harvested by scraping and cell pellets were frozen in liquid nitrogen and stored at −80°C until purification.

For purification, cells were resuspended in hypotonic buffer (15 mM KCl, 10 mM HEPES, 0.25 mM DTT, pH 7.5) and lysed by Dounce homogenization. The total membrane fraction was prepared, and membranes were resuspended in buffer A (0.2M NaCl, 0.1M KCl, 0.05M HEPES, 0.25 mM DTT, 4% sucrose, 1 mM ATP, 1 μM GBC, pH 7.5) and solubilized with 0.5% Digitonin. The soluble fraction was incubated with anti-FLAG M2 affinity agarose for 4 hr and eluted with buffer A (without sugar) containing 0.25 mg/mL FLAG peptide. Purified channels were concentrated to ~1–1.5 mg/mL and used immediately for cryo grid preparation.

## Sample preparation and data acquisition for cryo-EM analysis

In our previous data sets, most micrographs were of ice that was either too thin, which tended to exclude channel complex from the hole and induce highly preferred orientation, or of ice that was too thick, which gave good particle distribution and good angular coverage, but had lower contrast. Thus the current data set was the result of efforts to optimize ice thickness in order to retain high contrast and particle distribution. This was achieved through varying blotting time and also through extensive screening of the grid in order to find optimal regions. Two grids were imaged from the same purification and were prepared as follows: 3 μL of purified $K_{ATP}$ channel complex was loaded onto UltrAuFoil gold grids which had been glow-discharged for 60 s at 15 mA with a Pelco EasyGlow. The sample was blotted for 2 s (blot force −4; 100% humidity) and cryo-plunged into liquid ethane cooled by liquid nitrogen using a Vitrobot Mark III (FEI, Hillsboro, OR).

Single-particle cryo-EM data was collected on a Titan Krios 300 kV cryo-electron microscope (FEI) in the Multi-Scale Microscopy Core at Oregon Health and Science University, assisted by the automated acquisition program SerialEM. Images were recorded on the Gatan K2 Summit direct electron detector in super-resolution mode, post-GIF (20 eV window), at the nominal magnification 81,000x (calibrated image pixel-size of 1.720 Å; super-resolution pixel size 0.86 Å); defocus was varied

between −1.4 and −3.0 µm across the dataset (*Table 1*). The dose rate was kept around 2.7 e⁻/Å²/sec, with a frame rate of 4 frames/sec, and 60 frames in each movie, which gave a total dose of approximately 40 e⁻/Å². In total, 2180 movies were recorded.

## Image processing

The raw frame stacks were gain-normalized and then aligned and dose-compensated using Motioncor2 (*Zheng et al., 2017*) with patch-based alignment (5 × 5). CTF parameters were estimated from the aligned frame sums using CTFFIND4 (*Rohou and Grigorieff, 2015*). Particles were picked automatically using DoGPicker (*Voss et al., 2009*) with a broad threshold range in order to reduce bias. Subsequently, each image was analyzed manually in order to recover any particles missed by automatic picking and remove obviously bad micrographs from the data set. This resulted in ~250,000 raw particles as input for subsequent 2D classification using Relion-2 (*Kimanius et al., 2016*). After four rounds of 2D classification, ~160,000 particles remained in the data set, in which only classes displaying fully assembled complexes and high signal/noise were selected. These 160K particles were re-extracted at 1.72 Å/pixel and were used as input for 3D classification in Relion-2. Note only images collected in the current study were used for the 3D reconstruction described below.

Extensive 3D classification was performed in order to sample the heterogeneity within the data. Symmetry was not imposed at this step in order to select only the best four-fold symmetric classes. Up to 4 consecutive rounds of classification were performed, specifying 4 or 5 classes per round. Individual classes and combinations of classes were refined independently and lead to very similar structures. The two best classes from round 2 were combined (~63,000 particles), and then particles were re-extracted from super-resolution micrographs with a box size of 600 pixels. A soft mask encompassing the entire complex was used during refinement in Relion, with C4 symmetry imposed, which resulted in a 4.07 Å reconstruction using the gold-standard FSC cutoff (*Figure 1—figure supplement 1D*). These particle assignments were then imported into Frealign (*Grigorieff, 2016*) with the unbinned particle data and further classified and refined. To prevent overfitting, the resolution limit for every alignment iteration never exceeded the 0.9 value of the Frealign calculated FSC. The final round of refinement was done with an alignment limit of 4.8 Å, and the 0.143 value of the FSC was 3.63 Å (*Figure 1—figure supplements 1D* and *2C*). The masking in Frealign used the low-pass filtering (40 Å) and weighting (0.3) options to best minimize the effect of the micelle on alignment. A 'Score to Weight Constant' of 3.0 was used. Local resolution was calculated on unfiltered half maps with the Bsoft package, which showed the resolution was highest in the Kir.6.2/TMD0 core, as well as the SUR1 helices surrounding the GBC-binding pocket (between 3.3–3.7), and lowest in the NBDs and some of the external helices of TMD1/TMD2 of SUR1 (*Figure 1—figure supplement 2B*).

## Model building

In our previous reconstruction, many side-chains were left out of the final model as there was not sufficient density to support their placement (*Martin et al., 2017*). In the current reconstruction, there is good density for nearly every side chain of Kir6.2, TMD0, and the inner helices of the ABC core structure of SUR1. Thus using our previous structure as the starting template, we rebuilt nearly all of the structure with RosettaCM (*Song et al., 2013*), using the density as an additional constraint. This region included Kir6.2, TMD0/L0, and TMD1 and TMD2. The lowest energy models were very similar to one another, thus the lowest energy model was selected for each region. The resulting model was then minimized once in CNS (*Brünger et al., 1998*), substituting in the RSRef real-space target function (*Chapman et al., 2013*), adding ($\varphi,\psi$) backbone torsion angle restraints, and imposing noncrystallographic symmetry (NCS) constraints. In the density map, NBD1 and NBD2 showed signs of disorder, so our previously deposited NBD models were left as polyalanine chains and only refined as rigid bodies with RSRef. The distal N- and C-termini of Kir6.2, as well as the linker between NBD1 and TMD2 in SUR1 were not observed in the density map, and thus were left out the model. The final model contains residues 32–352 for Kir6.2, and residues 6–615 (TMD0/L0 + TMD1), 678–744 and 770–928 (NBD1), 1000–1044 and 1061–1319 (TMD2), and 1343–1577 (NBD2) for SUR1. All structure figures were produced with UCSF Chimera (*Pettersen et al., 2004*) and PyMol (http://www.pymol.org). Pore radius calculations were performed with HOLE (*Smart et al., 1996*).

## Sequence alignments

Multiple sequence alignment was performed with the T-Coffee server (*Notredame et al., 2000*). Output was saved in Clustal Aln format, and then imported and visualized in UCSF Chimera.

## Functional studies of GBC binding mutants

Point mutations were introduced into hamster SUR1 cDNA in pECE using the QuikChange site-directed mutagenesis kit (Stratagene, San Diego, CA). Mutations were confirmed by DNA sequencing. Mutant SUR1 cDNAs and rat Kir6.2 in pcDNA1 were co-transfected into COSm6 cells using FuGENE6, as described previously (*DevaTraneni et al., 2015*) and used for Western blotting, $^{86}$Rb$^+$ efflux assays, and electrophysiology as described below.

For Western blotting, cells were lysed in 20 mM HEPES, pH 7.0/5 mM EDTA/150 mM NaCl/1% Nonidet P-40 with cOmplete$^{TM}$ protease inhibitors (Roche, Basel, Switzerland) 48–72 hr post-transfection. Proteins in cell lysates were separated by SDS/PAGE (8%), transferred to nitrocellulose membrane, probed with rabbit anti-SUR1 antibodies against a C-terminal peptide of SUR1 (KDSVFASFVRADK), followed by HRP-conjugated anti-rabbit secondary antibodies (Amersham Pharmacia, Pittsburgh, PA), and visualized by chemiluminescence (Super Signal West Femto; Pierce, Rochford, IL) with FluorChem E (ProteinSimple, San Jose, CA).

For $^{86}$Rb$^+$ efflux assays, cells were plated and transfected in 12-well plates. Twenty-four to thirty-six hours post-transfection, cells were incubated overnight in medium containing $^{86}$RbCl (0.1 µCi/ml). The next day, cells were washed in Krebs-Ringer solution twice and incubated with metabolic inhibitors (2.5 µg/ml oligomycin and 1 mM 2-deoxy-D-glucose) in Krebs-Ringer solution for 30 min in the presence of $^{86}$Rb$^+$. Following two quick washes in Krebs-Ringer solutions containing metabolic inhibitors and 0.1% DMSO (vehicle control), 100 nM GBC, or 1 µM GBC, 0.5 ml of the same solution was added to each well. At the end of 2.5 min, efflux solution was collected for scintillation counting and new solution was added. The steps were repeated for 5, 7.5, 15, 25, and 40 min cumulative time points. After the 40 min time point efflux solution was collected, cells were lysed in Krebs-Ringer containing 1% SDS. $^{86}$Rb$^+$ in the solution and the cell lysate was counted. The percentage efflux was calculated as the radioactivity in the efflux solution divided by the total activity from the solution and cell lysate, as described previously (*Chen et al., 2013*; *Yan et al., 2007*). Note we used higher concentrations of GBC for these experiments than the electrophysiology experiments described below as in the latter the channels were exposed directly in isolated membrane patches to GBC, thus requiring lower concentrations. Experiments were repeated three-four times and for each experiment, untransfected cells were included as a negative control.

For electrophysiology experiments, cells co-transfected with SUR1 and Kir6.2 along with the cDNA for the green fluorescent protein GFP (to facilitate identification of transfected cells) were plated onto glass coverslips twenty-four hours after transfection and recordings made in the following two days. All experiments were performed at room temperature as previously described (*Devaraneni et al., 2015*). Micropipettes were pulled from non-heparinized Kimble glass (Fisher Scientific, Waltham, MA) on a horizontal puller (Sutter Instrument, Co., Novato, CA, USA). Electrode resistance was typically 1–2 MΩ when filled with K-INT solution containing 140 mM KCl, 10 mM K-HEPES, 1 mM K-EGTA, pH 7.3. ATP was added as the potassium salt. Inside-out patches of cells bathed in K-INT were voltage-clamped with an Axopatch 1D amplifier (Axon Inc., Foster City, CA). ATP (as the potassium salt) or GBC at 10 nM or 100 nM were added to K-INT as specified in the figure legend. All currents were measured at a membrane potential of −50 mV (pipette voltage =+50 mV). Data were analyzed using pCLAMP10 software (Axon Instrument, Sunnyvale, CA). Off-line analysis was performed using Microsoft Excel programs. Data were presented as mean ±standard error of the mean (s.e.m).

## Data resources

The accession numbers for the structure presented in this paper are PDB: 6BAA and EMD: EMD-7073.

## Acknowledgements

The INS-1 cell clone 832/13 was kindly provided by Dr. Christopher Newgard. We thank Emily Rex and Zhongying Yang for technical assistance, and Dr. Matt Whorton, Dr. Dale Fortin, and Veronica Cochrane for critical readings of the manuscript. We also thank the staff at the Multiscale Microscopy Core (MMC) of Oregon Health and Science University (OHSU), the OHSU-FEI living lab and Intel for technical support. This work was supported by the National Institutes of Health grants R01DK066485 (to S-L S) and F31DK105800 (to GMM).

## Additional information

### Funding

| Funder | Grant reference number | Author |
|---|---|---|
| National Institute of Diabetes and Digestive and Kidney Diseases | R01DK066485 | Show-Ling Shyng |
| National Institute of Diabetes and Digestive and Kidney Diseases | F31DK105800 | Gregory M Martin |

The funders had no role in study design, data collection and interpretation, or the decision to submit the work for publication.

### Author contributions

Gregory M Martin, Conceptualization, Data curation, Formal analysis, Funding acquisition, Validation, Investigation, Visualization, Methodology, Writing—original draft, Writing—review and editing; Balamurugan Kandasamy, Data curation, Formal analysis, Investigation, Writing—review and editing; Frank DiMaio, Software, Validation, Methodology, Writing—review and editing; Craig Yoshioka, Data curation, Software, Formal analysis, Supervision, Validation, Investigation, Visualization, Writing—review and editing; Show-Ling Shyng, Conceptualization, Resources, Data curation, Formal analysis, Supervision, Funding acquisition, Validation, Investigation, Visualization, Writing—original draft, Project administration, Writing—review and editing

### Author ORCIDs

Frank DiMaio http://orcid.org/0000-0002-7524-8938
Show-Ling Shyng http://orcid.org/0000-0002-8230-8820

### Decision letter and Author response

Decision letter https://doi.org/10.7554/eLife.31054.028
Author response https://doi.org/10.7554/eLife.31054.029

## Additional files

### Supplementary files

• Transparent reporting form
DOI: https://doi.org/10.7554/eLife.31054.021

### Major datasets

The following datasets were generated:

| Author(s) | Year | Dataset title | Dataset URL | Database, license, and accessibility information |
|---|---|---|---|---|
| Shyng SL, Yoshioka C, Martin G | 2017 | Cryo-EM structure of the pancreatic beta-cell KATP channel bound to ATP and glibenclamide | http://www.ebi.ac.uk/pdbe/entry/emdb/EMD-7073 | Publicly available at the EMDataBank (accession no. EMD-7073) |

| Shyng SL, Yoshioka C, Martin G | 2017 | Cryo-EM structure of the pancreatic beta-cell KATP channel bound to ATP and glibenclamide | http://www.rcsb.org/pdb/search/structid-Search.do?structureId=6BAA | Publicly available at RCSB Protein Data Bank (accession no. 6BAA) |

The following previously published dataset was used:

| Author(s) | Year | Dataset title | Dataset URL | Database, license, and accessibility information |
|---|---|---|---|---|
| Martin GM, Yoshioka C, Rex EA, Fay JF, Xie Q, Whorton MR, Chen JZ, Shyng SL | 2017 | Cryo-EM structure of the pancreatic ATP-sensitive K+ channel SUR1/Kir6.2 in the presence of ATP and glibenclamide | https://www.rcsb.org/pdb/explore/explore.do?structureId=5TWV | Publicly available at RCSB Protein Data Bank (accession no: 5TWV) |

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
