## [Decision Letter]

Thank you for submitting your article "Anti-diabetic drug binding site in K<sub>ATP</sub> channels revealed by Cryo-EM" for consideration by *eLife*. Your article has been reviewed by three peer reviewers, and the evaluation has been overseen by Kenton Swartz as the Reviewing Editor and Richard Aldrich as the Senior Editor. The following individuals involved in review of your submission have agreed to reveal their identity: John L Rubinstein (Reviewer #1); Colin G. Nichols (Reviewer #2); Cristina Paulino (Reviewer #3).

The reviewers have discussed the reviews with one another and the Reviewing Editor has drafted this decision to help you prepare a revised submission.

Summary:

The manuscript by Martin et al. report the 3.8 Å cryo-EM structure of the pancreatic KATP channel. This chimeric complex is composed of the potassium channel Kir6.2 and the ABC transporter unit SUR1, which itself shows no transport activity but functions as a regulator. The cryo-EM density is of very good quality and allowed the authors to build a model with high confidence for the majority of the protein. The structure reveals a detailed atomic interpretation for the binding site of the inhibitors ATP and glibenclamide, which the authors use to disclose a mechanism of inhibition. The results are complemented with comprehensive functional data, which strengthen the novel findings. While the overall architecture of this protein complex was revealed by two independent groups earlier this year, including the authors themselves, the higher-resolution data presented in this manuscript provides unique and exciting new insights. In light of the high pharmacological relevance of the protein and its implication in diabetes, these new findings will be of interest to several research fields and with no doubt have a big impact. In conclusion, the manuscript fits the high standards of *eLife*. The following are suggestions for improving the manuscript.

1) We understand that the field is highly competitive and that the publication of an intermediate resolution map was of great value. However, in light of the short time between the previous and current manuscript, could the authors specify whether all the micrographs used for reconstruction in this publication have been newly acquired, or whether the old data set was also included? Also, it is unclear from the manuscript how the resolution was improved with the same sample. Some space should be dedicated to answering this question.

2) The authors are likely aware of the recently reported bug for the Relion versions 2.0.4, 2.0.6 and 2.1b0, where the inclusion of reference-bias may have led to the over-fitting of the refined maps. Could the authors confirm that the final reconstruction was calculated with the newest Relion version 2.1b1, such that the indicated resolution indeed refers to "gold-standard" conditions?

3) Can the authors confirm that the final map was indeed calculated from the super-resolution data with a pixel size of 0.86 Å/pixel? Table 1 rather suggest that the reconstruction was done with the binned data at a pixel size of 1.72 Å/pixel. However, a resolution of 3.8 Å would then correspond to over 90% of Nyquist, which is more than one would expect to be possible.

4) We strongly suggest the authors include a validation for the model refinement, by plotting the FSC curve for the FSCsum, FSCwork and FSCfree, which correspond to: FSC between map and model; between shaken-model refined against halfmap1 and halfmap1; and between shaken-model refined against halfmap1 and halfmap2, respectively. We also request that you deposit the mask used to determine the final resolution. Although this is not yet mandatory it is highly recommended, as was discussed during the last 3DEM-GRC to improve standards in cryo-EM.

5) In the Results section, the authors write "this part of the map was almost entirely at or below 3.7 Å" The statement may be misleading because (1) local resolution determination is not typically quantitative and (2) the figures were presumably rendered at the 3.8 Å resolution, not the 3.7 Å resolution (so a reader would not benefit from the improved local resolution).

6) The current resolution is not sufficient to define hydrogen bond between atoms, therefore it is suggested to remove the dotted lines in all the figures.

7) In subsection “The ATP binding site and mechanism of ATP inhibition” the authors mention that the structure provides details of the intricate relationship between ATP and PIP2 regulation, and refers to Figure 5 (which is not mentioned in the figure legend). However, the structure illustrates the ATP binding site but does not inform the PIP2 site. Since PIP2 is (presumably) not resolved in the structure, then the nature of PIP2 versus ATP interaction is mere speculation. Work by Envetchakul et al., (2000) showed that ATP and PIP2 act essentially as mutually exclusive ligands, binding preferentially to the unliganded closed, or unliganded open channel, respectively, which is not trivially consistent with the cartoon in Figure 8. The following paragraph (in subsection “The ATP binding site and mechanism of ATP inhibition”) speculates that bound ATP acts 'like a stopper that prevents movement of the Kir6.2 N-terminus relative to L0 that is necessary to open the channel'. Subsequent work by Enkvetchakul et al., (2001) and Li et al., (2000) showed that ATP can interact with the open channel directly, confirming the allosteric nature of ATP inhibition, essentially obviating an obligate gating step that the authors propose.

8) Similarly, in subsection “Mechanistic insights of GBC binding and inhibition” the authors 'propose' how GBC binding controls SUR1 conformation and gating, but this is mere speculation and should be stated as 'hypothesize'.

9) At several places in the text, assertions are made without references given. This should be carefully corrected. There are several typos and grammatical oddities. Rather than enumerate them individually, a hand marked copy of the text is attached to help the authors.

---

## [Author Response]

Summary:The manuscript by Martin et al. report the 3.8 Å cryo-EM structure of the pancreatic KATP channel. This chimeric complex is composed of the potassium channel Kir6.2 and the ABC transporter unit SUR1, which itself shows no transport activity but functions as a regulator. The cryo-EM density is of very good quality and allowed the authors to build a model with high confidence for the majority of the protein. The structure reveals a detailed atomic interpretation for the binding site of the inhibitors ATP and glibenclamide, which the authors use to disclose a mechanism of inhibition. The results are complemented with comprehensive functional data, which strengthen the novel findings. While the overall architecture of this protein complex was revealed by two independent groups earlier this year, including the authors themselves, the higher-resolution data presented in this manuscript provides unique and exciting new insights. In light of the high pharmacological relevance of the protein and its implication in diabetes, these new findings will be of interest to several research fields and with no doubt have a big impact. In conclusion, the manuscript fits the high standards of eLife. The following are suggestions for improving the manuscript.1) We understand that the field is highly competitive and that the publication of an intermediate resolution map was of great value. However, in light of the short time between the previous and current manuscript, could the authors specify whether all the micrographs used for reconstruction in this publication have been newly acquired, or whether the old data set was also included? Also, it is unclear from the manuscript how the resolution was improved with the same sample. Some space should be dedicated to answering this question.

As stated in Materials and methods section, we collected more data and optimized the grid preparation conditions, in particular ice thickness and particle distributions to improve resolution. We also clarified that for the structure presented in this paper, we did not include data collected for the previous publication (Martin et al., 2017).

2) The authors are likely aware of the recently reported bug for the Relion versions 2.0.4, 2.0.6 and 2.1b0, where the inclusion of reference-bias may have led to the over-fitting of the refined maps. Could the authors confirm that the final reconstruction was calculated with the newest Relion version 2.1b1, such that the indicated resolution indeed refers to "gold-standard" conditions?

We thank the reviewers for pointing this out. The Relion bug was reported shortly after we submitted our paper. To address this, we had already started reprocessing the data using Relion version 2.1b1 upon its release. In addition, we were also starting to use cisTEM (Frealign) to further improve the maps via 3D classification and improved masking of the micelle. This combination resulted in a map that was almost the same, though slightly improved, from the one previously obtained from the buggy version of Relion. We have now revised the Materials and methods section to reflect these changes.

3) Can the authors confirm that the final map was indeed calculated from the super-resolution data with a pixel size of 0.86 Å/pixel? Table 1 rather suggest that the reconstruction was done with the binned data at a pixel size of 1.72 Å/pixel. However, a resolution of 3.8 Å would then correspond to over 90% of Nyquist, which is more than one would expect to be possible.

The initial reconstruction was done with the data binned using Fourier cropping to a pixel size of 1.72 Å/pixel. We have re-processed using the unbinned data, and obtained an EM map of similar resolution (even slightly improved, ~3.8Å to ~3.6Å)

4) We strongly suggest the authors include a validation for the model refinement, by plotting the FSC curve for the FSCsum, FSCwork and FSCfree, which correspond to: FSC between map and model; between shaken-model refined against halfmap1 and halfmap1; and between shaken-model refined against halfmap1 and halfmap2, respectively. We also request that you deposit the mask used to determine the final resolution. Although this is not yet mandatory it is highly recommended, as was discussed during the last 3DEM-GRC to improve standards in cryo-EM.

We have now included the FSC curve for the FSCsum, FSCwork and FSCfree as requested to provide validation for the model refinement. We have also deposited the mask used to determine the final resolution as requested.

5) In the Results section, the authors write "this part of the map was almost entirely at or below 3.7 Å" The statement may be misleading because (1) local resolution determination is not typically quantitative and (2) the figures were presumably rendered at the 3.8 Å resolution, not the 3.7 Å resolution (so a reader would not benefit from the improved local resolution).

We have revised the text according to the reviewers’ suggestion.

6) The current resolution is not sufficient to define hydrogen bond between atoms, therefore it is suggested to remove the dotted lines in all the figures.

The dotted lines were meant to indicate potential van der Waals or electrostatic interactions between two residues as stated in Figure 3 legend rather than actual hydrogen bonds observed. We have now also included this clarification for all figures that contain dotted lines.

7) In subsection “The ATP binding site and mechanism of ATP inhibition” the authors mention that the structure provides details of the intricate relationship between ATP and PIP2 regulation, and refers to Figure 5 (which is not mentioned in the figure legend). However, the structure illustrates the ATP binding site but does not inform the PIP2 site. Since PIP2 is (presumably) not resolved in the structure, then the nature of PIP2 versus ATP interaction is mere speculation. Work by Envetchakul et al., (2000) showed that ATP and PIP2 act essentially as mutually exclusive ligands, binding preferentially to the unliganded closed, or unliganded open channel, respectively, which is not trivially consistent with the cartoon in Figure 8. The following paragraph (in subsection “The ATP binding site and mechanism of ATP inhibition”) speculates that bound ATP acts 'like a stopper that prevents movement of the Kir6.2 N-terminus relative to L0 that is necessary to open the channel'. Subsequent work by Enkvetchakul et al., (2001) and Li et al., (2000) showed that ATP can interact with the open channel directly, confirming the allosteric nature of ATP inhibition, essentially obviating an obligate gating step that the authors propose.

We apologize for the oversight of the missing Figure 5 legend, which is now added. To address the reviewers’ concerns, we have revised the text and model figure/legend (now Figure 9) regarding ATP and PIP_2_ regulation to make clear the hypothetical nature of the model and discussion, citing previous studies by others that are relevant to the discussion.

8) Similarly, in subsection “Mechanistic insights of GBC binding and inhibition” the authors 'propose' how GBC binding controls SUR1 conformation and gating, but this is mere speculation and should be stated as 'hypothesize'.

Again, we have reworded the text and legend as suggested.

9) At several places in the text, assertions are made without references given. This should be carefully corrected. There are several typos and grammatical oddities. Rather than enumerate them individually, a hand marked copy of the text is attached to help the authors.

We have gone through the text carefully to include references where needed and to correct grammatical errors.